# MOFFLOW: FLOW MATCHING FOR STRUCTURE PREDICTION OF METAL-ORGANIC FRAMEWORKS

**Nayoung Kim**[1][*]    **Seongsu Kim**[2]    **Minsu Kim**[1]    **Jinkyoo Park**[1]    **Sungsoo Ahn**[1]

[1]Korea Advanced Institute of Science and Technology (KAIST)
[2]Pohang University of Science and Technology (POSTECH)
{nayoungkim, min-su, jinkyoo.park, sungsoo.ahn}@kaist.ac.kr
{seongsukim}@postech.ac.kr

## ABSTRACT

Metal-organic frameworks (MOFs) are a class of crystalline materials with promising applications in many areas such as carbon capture and drug delivery. In this work, we introduce MOFFLOW, the first deep generative model tailored for MOF structure prediction. Existing approaches, including *ab initio* calculations and even deep generative models, struggle with the complexity of MOF structures due to the large number of atoms in the unit cells. To address this limitation, we propose a novel Riemannian flow matching framework that reduces the dimensionality of the problem by treating the metal nodes and organic linkers as rigid bodies, capitalizing on the inherent modularity of MOFs. By operating in the $SE(3)$ space, MOFFLOW effectively captures the roto-translational dynamics of these rigid components in a scalable way. Our experiment demonstrates that MOFFLOW accurately predicts MOF structures containing several hundred atoms, significantly outperforming conventional methods and state-of-the-art machine learning baselines while being much faster. Code available at https://github.com/nayoung10/MOFFlow.

## 1 INTRODUCTION

Metal-organic frameworks (MOFs) are a class of crystalline materials that have recently received significant attention for their broad range of applications, including gas storage (Li et al., 2018), gas separations (Qian et al., 2020), catalysis (Lee et al., 2009), drug delivery (Horcajada et al., 2012), sensing (Kreno et al., 2012), and water purification (Haque et al., 2011). They are particularly valued for their permanent porosity, high stability, and remarkable versatility due to their tunable structures. In particular, MOFs are tunable by adjusting their building blocks, i.e., metal nodes and organic linkers, to modify pore size, shape, and chemical characteristics to suit specific applications (Wang et al., 2013). Consequently, there is a growing interest in developing automated approaches to designing and simulating MOFs using computational algorithms.

Crystal structure prediction (CSP) is a task of central importance for automated MOF design and simulation. The importance of this task lies in the fact that the important functions of MOFs, such as pore size, surface area, and stability, are directly dependent on the crystal structure. The conventional approach to general CSP is based heavily on *ab initio* calculations using density functional theory (DFT; Kohn & Sham, 1965), often combined with optimization algorithms such as random search (Pickard & Needs, 2011) and Bayesian optimization (Yamashita et al., 2018) to iteratively explore the energy landscape. However, the reliance on DFT for such computations can be computationally expensive, especially for large and complex systems such as MOFs.

Deep generative models are promising solutions to accelerate the prediction of the MOF structure. Especially, diffusion models (Ho et al., 2020) and flow-based models (Lipman et al., 2023) have shown success in similar molecular structure prediction problems, e.g., small molecules (Xu et al., 2022; Jing et al., 2022), folded proteins (Jing et al., 2024; Lin et al., 2023), protein-ligand complex (Corso et al., 2023), and general crystals without the building block constraint (Gebauer et al.,

---

*Work partially done as a researcher at POSTECH.

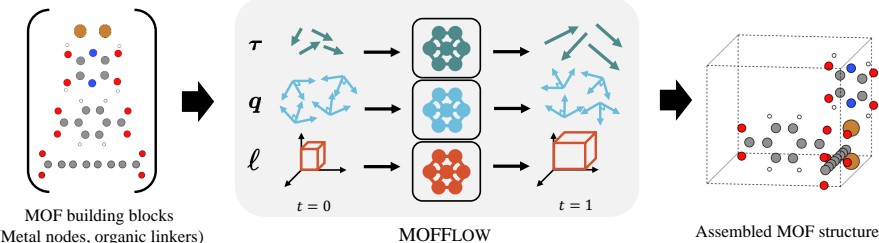

MOF building blocks
(Metal nodes, organic linkers)                MOFFLOW                Assembled MOF structure

Figure 1: **Overview of MOFFLOW.** MOFFLOW is a continuous normalizing flow that exploits the modular nature of MOFs by modeling the building blocks (i.e., metal nodes and organic linkers) as rigid bodies. It learns the vector fields for rotation ($q$), translation ($\tau$), and the lattice ($\ell$) that assembles the building blocks into a complete MOF structure.

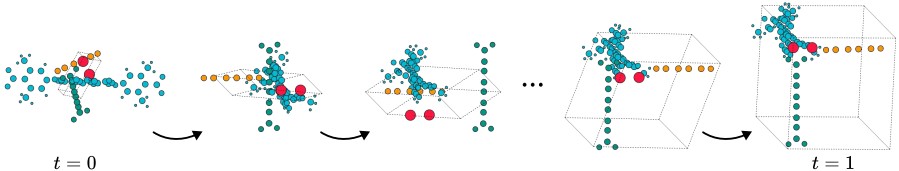

Figure 2: **Inference trajectory of MOFFLOW.** Visualization of the inference trajectory of MOF-FLOW from $t = 0$ to $t = 1$, showing the progressive assembly of building blocks. We wrap building block centroids inside the lattice for visual clarity.

2022; Xie et al., 2022; Jiao et al., 2024a; Miller et al., 2024). These models iteratively denoise the random structure using neural networks that act similar to force fields guiding atom positions toward a minimum energy configuration.

**Contribution.** In this work, we introduce MOFFLOW, the first deep generative model tailored for MOF structure prediction. MOFFLOW leverages the modular nature of MOFs, which can be decomposed into metal nodes and organic linkers (Figures 1 and 2). This decomposition enables us to design a generative model that predicts the roto-translation of these building blocks to match the ground truth structure. To achieve this, based on Riemannian flow matching (Chen & Lipman, 2024), we propose a new framework that generates rotations, translations, and lattice structure of the building blocks. We design the underlying neural network as a composition of building block encoder parameterized by an equivariant graph neural network based on a new attention module for encoding roto-translations and lattice parameters of the MOF.

We note that our method competes with existing deep-generative models (Gebauer et al., 2022; Xie et al., 2022; Jiao et al., 2024a; Miller et al., 2024) for general CSP that encompass MOFs as special members. However, our method is specialized for MOF structure prediction by exploiting the domain knowledge that the local structures of the MOF building blocks are shared across different MOF structures. This is particularly useful for reducing the large search space of MOF structures; We consider MOFs up to 2,200 atoms per unit cell (Boyd et al., 2019), whereas crystals for general CSP consist of up to 52 atoms per unit cell (Jain et al., 2013; Jiao et al., 2024a). In fact, as confirmed in our experiments, the recent deep generative model (Jiao et al., 2024a) for general CSP fails to scale to the large system size of MOFs. This also aligns with how torsional diffusion (Jing et al., 2022) improved over existing molecular conformer generation algorithms (Xu et al., 2021; 2022) by eliminating the redundant degree of freedom.

We benchmark our algorithm with the MOF dataset compiled by Boyd et al. (2019), consisting of 324,426 structures. We compare with conventional and deep learning-based algorithms for crystal structure prediction. Notably, MOFFLOW achieves a match rate of 31.69% on unseen MOF structures, whereas existing methods, despite being more computationally expensive, barely match any. We also demonstrate that MOFFLOW captures key MOF properties and scales efficiently to structures containing hundreds of atoms.

## 2 RELATED WORK

**Crystal structure prediction (CSP).** Traditional approaches to CSP rely on density functional theory (DFT) to identify energetically stable structures. To generate candidate structures, heuristic techniques such as random sampling (Pickard & Needs, 2011) and simple substitution rules (Wang et al., 2021) have been employed, alongside more sophisticated optimization algorithms such as Bayesian optimization (Yamashita et al., 2018), genetic algorithms (Yamashita et al., 2022), and particle swarm optimization (Wang et al., 2010). To address the computational burden with DFT calculations, many studies used machine learning as surrogates for energy evaluation (Jacobsen et al., 2018; Podryabinkin et al., 2019; Cheng et al., 2022).

Recently, deep generative models have emerged as a promising alternative to optimization-based methods (Court et al., 2020; Hoffmann et al., 2019; Noh et al., 2019; Yang et al., 2021; Hu et al., 2020; 2021; Kim et al., 2020; Ren et al., 2022). Notably, Jiao et al. (2024a) proposes an equivariant diffusion-based model to capture the periodic $E(3)$-invariance of crystal structure distributions, while Lin et al. (2024) and Jiao et al. (2024b) additionally consider lattice permutations and space group constraints, respectively. Miller et al. (2024) uses Riemannian flow matching to generate high-quality samples with fewer integration steps. However, these methods face significant challenges in predicting the MOFs structures, which often consist of hundreds of atoms per unit cell.

**MOF structure prediction.** Unlike general CSP, where a variety of algorithms have been developed, MOF structure prediction remains a significant challenge. Conventional MOF structure prediction methods heavily rely on predefined topologies to connect MOF building blocks (Marleny Rodriguez-Albelo et al., 2009; Wu & Jiang, 2024), restricting the discovery of structures with new topologies. To address this limitation, Darby et al. (2020) proposes to combine *ab initio* random structure searching (AIRSS; Pickard & Needs, 2011) with the Wyckoff alignment of molecules (WAM) method; however, the reliance on AIRSS makes it computationally expensive. We also note that a recent work (Fu et al., 2023) considered a related, yet different problem of MOF generation based on a deep generative model which does not include the structure generation.

**Flow matching.** Flow matching is a simulation-free approach for training continuous normalizing flows (Lipman et al., 2023; Albergo & Vanden-Eijnden, 2023; Liu et al., 2023). Since its introduction, various extensions have been proposed, such as generalization to Riemannian manifolds (Chen & Lipman, 2024) and efficiency improvements through optimal transport (Tong et al., 2024; Pooladian et al., 2023). Due to its flexibility and computational efficiency, flow matching has made notable progress in several related domains, including protein generation (Yim et al., 2023a;b; 2024; Bose et al., 2024), molecular conformation generation (Song et al., 2024), and CSP (Miller et al., 2024).

## 3 PRELIMINARIES

### 3.1 REPRESENTATION OF MOF STRUCTURES

**MOF representation.** The 3D crystal structure of a MOF can be represented with the periodic arrangement of the smallest repeating unit called the unit cell. A unit cell containing $N$ atoms can be represented with the tuple $\mathcal{S} = (\boldsymbol{X}, \boldsymbol{a}, \boldsymbol{\ell})$ where $\boldsymbol{X} = [x_n]_{n=1}^{N} \in \mathbb{R}^{N \times 3}$ is the atom coordinates, $\boldsymbol{a} = [a_n]_{n=1}^{N} \in \mathcal{A}^N$ is the atom types with $\mathcal{A}$ denoting the set of possible elements, and $\boldsymbol{\ell} = (a, b, c, \alpha, \beta, \gamma) \in \mathbb{R}_+^3 \times [0, 180]^3$ is the lattice parameter that describes the periodicity of the structure (Miller et al., 2024; Luo et al., 2024). In particular, the lattice parameter $\boldsymbol{\ell}$ can be transformed into a standard lattice matrix $L = (\boldsymbol{l}_1, \boldsymbol{l}_2, \boldsymbol{l}_3) \in \mathbb{R}^{3 \times 3}$, which defines the infinite crystal structure:

$$\{(a'_n, x'_n) | a'_n = a_n, x'_n = x_n + kL^\top, k \in \mathbb{Z}^{1 \times 3}\} \tag{1}$$

where $k = (k_1, k_2, k_3)$ is the set of integers representing the periodic translation of the unit cell.

**Block-wise representation of MOFs.** Here we introduce the blockwise representation of MOFs that decompose a given unit cell into constituent building blocks, i.e., the metal nodes and organic linkers. The blockwise representation is a tuple $\mathcal{S} = (\mathcal{B}, \boldsymbol{q}, \boldsymbol{\tau}, \boldsymbol{\ell})$ where $\mathcal{B} = \{\mathcal{C}_m\}_{m=1}^{M}$ corresponds to $M$ building blocks and $\boldsymbol{q} = [q_m]_{m=1}^{M}, \boldsymbol{\tau} = [\tau_m]_{m=1}^{M}$ corresponds to set of $M$ building block roto-translations $(q_m, \tau_m) \in SE(3)$. Moreover, each block $\mathcal{C}_m = (\boldsymbol{a}_m, \boldsymbol{Y}_m)$ has $N_m$ atoms with atom types $\boldsymbol{a}_m = [a_n]_{n=1}^{N_m} \in \mathcal{A}^{N_m}$ and local coordinates $\boldsymbol{Y}_m = [y_n]_{n=1}^{N_m} \in \mathbb{R}^{N_m \times 3}$ (defined

in Section 4.1). Our main assumption is that the building blocks can be composed by the roto-translations to form the MOF structure, i.e., the atomwise representation $(\boldsymbol{X}, \boldsymbol{a}, \boldsymbol{\ell})$ can be expressed by the blockwise representation $(\boldsymbol{q}, \boldsymbol{\tau}, \mathcal{B}, \boldsymbol{\ell})$ as $\boldsymbol{X} = \mathrm{Concat}(\boldsymbol{X}_1, \dots, \boldsymbol{X}_M)$, $\boldsymbol{X}_m = (q_m, \tau_m) \cdot \boldsymbol{Y}_m$, where $\boldsymbol{X}_m$ is the result of the roto-translated local coordinate represented by group action $\cdot$. We express the global coordinate $\boldsymbol{X}$ as the concatenation $\boldsymbol{X}_m$'s without loss of generality.

## 3.2 FLOW MATCHING ON RIEMANNIAN MANIFOLDS

Flow matching is a method to train continuous normalizing flow (CNF; Chen et al., 2018) without expensive ordinary differential equation (ODE) simulations (Lipman et al., 2023). Here, we introduce the flow matching generalized to Riemannian manifolds (Chen & Lipman, 2024).

**CNF in Riemannian manifold.** We consider a smooth and connected Riemann manifold $\mathcal{M}$ with metric $g$ where each point $x \in \mathcal{M}$ is associated with tangent space $\mathcal{T}_x\mathcal{M}$ and inner product $\langle \cdot, \cdot \rangle_g$. We consider learning a CNF $\phi_t : \mathcal{M} \to \mathcal{M}$ defined with the ODE $\frac{d}{dt}\phi_t(x) = u_t(\phi_t(x))$ where $\phi_0(x) = x$ and $u_t(x) \in \mathcal{T}_x\mathcal{M}$ is the time-dependent smooth vector field. The vector field $u_t(x)$ transforms a prior distribution $p_0$ to $p_t$ according to the following push-forward equation:

$$p_t(x) = [\phi_t]_* p_0(x) - p_0(\phi_t^{-1}(x)) \exp\left(-\int_0^t \nabla \cdot u_t(x_s)ds\right), \qquad t \in [0, 1], \qquad (2)$$

where $\nabla \cdot$ is the divergence operator and $x_s = \phi_s(\phi_t^{-1}(x))$.

**Conditional flow matching on Riemannian manifold.** The goal of CNF is to learn a vector field $u_t(\cdot)$ that transforms a simple prior distribution $p_0$ so that $p_1$ closely approximates some target distribution $q$. Given a vector field $u_t(\cdot)$ and the corresponding probability paths $\{p_t\}_{t\in[0,1]}$, one can train a neural network $v_t(x; \theta)$ with the flow matching objective $\mathcal{L}_{\mathrm{FM}}(\theta) = \mathbb{E}_{t,p_t(x)}[\|v_t(x; \theta) - u_t(x)\|_g^2]$, where $t \sim \mathcal{U}[0, 1]$, $x \sim p_t(x)$, and $\|\cdot\|_g$ is the norm induced by the metric $g$. However, the flow matching objective lacks analytic form of $u_t(x)$ that transforms the prior $p_0$ into the target $q$. The key insight of conditional flow matching objective is to instead learn a conditional vector field $u_t(x|x_1)$ for a data point $x_1$ defined as $\mathcal{L}_{\mathrm{CFM}}(\theta) = \mathbb{E}_{t,p_1(x),p_t(x|x_1)}[\|v_t(x; \theta) - u_t(x|x_1)\|_g^2]$, where we let $p_0(x|x_1) = p_0(x)$ and $p_1(x|x_1) \approx \delta(x - x_1)$ with $\delta(\cdot)$ being the Dirac distribution. The key idea of conditional flow matching is that one can derive the conditional vector field $u_t(x|x_1)$ that marginalizes over data points $x_1 \sim q$ accordingly to induce the vector field $u_t(x)$ transforming the prior $p_0$ into the desired distribution $q$. To construct $u_t(x|x_1)$, Chen & Lipman (2024) proposes defining conditional flow $x_t = \phi_t(x_0|x_1)$ from the geodesic path (minimum length curve) connecting two points $x_0, x_1 \in \mathcal{M}$ by $x_t = \exp_{x_0}(t \log_{x_0}(x_1))$ for $t \in [0, 1]$, where $\exp_x$ and $\log_x$ are the exponential and logarithmic map at point $x \in \mathcal{M}$, respectively. Then the desired conditional vector field can be derived as the time derivative, i.e., $u(x_t|x_1) = \frac{d}{dt}\phi_t(x_0|x_1)$.

At optimum, $v_\theta$ generates $p_t^\theta = p_t$ with starting point $p_0^\theta = p_0$ and end point $p_1^\theta = p_1$. At inference, we sample from the prior $p_0^\theta$ and propagate $t$ from 0 to 1 using one of the existing ODE solvers. Note that each training step is faster than the methods based on adjoint sensitivity (Chen et al., 2018) since conditional flow matching does not require solving the ODE defined by the neural network.

## 4 METHODS

In this section, we introduce MOFFLOW, a novel approach for MOF structure prediction based on the rigid-body roto-translation of building blocks to express the global atomic coordinates. To this end, using the Riemannian flow matching framework, we learn a CNF $p_\theta(\boldsymbol{q}, \boldsymbol{\tau}, \boldsymbol{\ell}|\mathcal{B})$ that predicts the blockwise roto-translations and the lattice parameter from the given building blocks. Compared to conventional CSP approaches defined on atomic coordinates (Jiao et al., 2024a;b; Lin et al., 2024; Miller et al., 2024), MOFFLOW enjoys the reduced search space, i.e., the dimensionality of blockwise roto-translation and the atomic coordinates are $6M$ and $3N$, respectively ($6M \ll 3N$).

### 4.1 CONSTRUCTION OF LOCAL COORDINATES

To incorporate the MOF symmetries, we devise a scheme to consistently define the local coordinate $\boldsymbol{Y}$ regardless of the initial pose of the building block. Given a building block $\mathcal{C} = (\boldsymbol{a}, \boldsymbol{Y})$, our

goal is to define a global-to-local function $f : \mathcal{A} \times \mathbb{R}^{N \times 3} \to \mathbb{R}^{N \times 3}$ that defines a consistent local coordinate system; that is, the function should satisfy

$$f(\boldsymbol{a}, \boldsymbol{X}Q^\top + 1_N t^\top) = f(\boldsymbol{a}, \boldsymbol{X}), \quad \forall Q \in SO(3), t \in \mathbb{R}^3. \tag{3}$$

where $\boldsymbol{X}$ is a random conformation of the building block. We can then define $\boldsymbol{Y} = f(\boldsymbol{a}, \boldsymbol{X})$. Such property can be satisfied by a composition of (1) translation by subtracting the centroid and (2) rotation by aligning with the principal component analysis (PCA) axes:

$$f(a, \boldsymbol{X}) = C(\boldsymbol{X})\mathcal{R}(\boldsymbol{a}, C(\boldsymbol{X})), \quad C(\boldsymbol{X}) = \boldsymbol{X} - 1_N \left( \frac{1}{N} \sum_{n=1}^{N} x_n^\top \right). \tag{4}$$

Here, $C(\boldsymbol{X})$ denotes subtraction of the centroid and $\mathcal{R}(\boldsymbol{a}, \boldsymbol{X})$ denotes rotation to align the building block with the PCA axes whose sign is fixed by a reference vector, following Gao & Günnemann (2022). See Appendix E for more details.

## 4.2 FLOW MATCHING FOR MOF STRUCTURE PREDICTION

In this section, we present our approach to training the generative model $p_\theta(\boldsymbol{q}, \boldsymbol{\tau}, \boldsymbol{\ell}|\mathcal{B})$ using the flow matching framework. We first explain how our method ensures $SE(3)$-invariance and introduce a metric for independent treatment of the components $\boldsymbol{q}$, $\boldsymbol{\tau}$, and $\boldsymbol{\ell}$. Next, we outline the key elements for flow matching – i.e., the definition of priors, conditional flows, and the training objective.

To preserve crystal symmetries, we design the framework such that the generative model is invariant to rotation, translation, and permutation of atoms and building blocks. Rotation invariance is guaranteed by using the rotation-invariant lattice parameter representation and canonicalizing the atomic coordinates based on the standard lattice matrix (Miller et al., 2024; Luo et al., 2024). Translation invariance is achieved by operating on the mean-free system where the building blocks are centered at the origin – i.e., $\frac{1}{M} \sum_{m=1}^{M} \tau_m = 0$. This is the only way to ensure translation invariance on $SE(3)^M$ as no $\mathbb{R}^3$ invariant probability measure exists (Yim et al., 2023b). Permutation invariance is addressed using equivariant graph neural networks (Satorras et al., 2021) and Transformers (Vaswani, 2017) as the backbone.

**Metric for $SE(3)$.** Following Yim et al. (2023b), we treat $SO(3)$ and $\mathbb{R}^3$ independently by defining an additive metric on $SE(3)$ as $\langle (q, \tau), (q', \tau') \rangle_{SE(3)} = \langle q, q' \rangle_{SO(3)} + \langle \tau, \tau' \rangle_{\mathbb{R}^3}$. Here, $\langle q, q' \rangle_{SO(3)}$ and $\langle \tau, \tau' \rangle_{\mathbb{R}^3}$ are inner products defined as $\langle q, q' \rangle_{SO(3)} = \operatorname{tr}(qq'^\top)/2$ and $\langle \tau, \tau' \rangle_{\mathbb{R}^3} = \tau^\top \tau'$ for $q, q' \in \mathfrak{so}(3)$ with $\mathfrak{so}(3)$ denoting the Lie algebra of $SO(3)$ and $\tau, \tau' \in \mathbb{R}^3$.

**Priors.** For each rotation $q$ and translation $\tau$, the prior are chosen as the uniform distribution on $SO(3)$ and standard normal distribution on $\mathbb{R}^3$, respectively. For the lattice parameter $\boldsymbol{\ell}$, we follow Miller et al. (2024) and use log-normal and uniform distributions. Specifically, for the lengths, we let $p_0(a, b, c) = \prod_{\lambda \in \{a,b,c\}} \text{LogNormal}(\lambda; \mu_\lambda, \sigma_\lambda)$ where the parameters are learned with the maximum-likelihood objective (Appendix C). For the angles, we use Niggli reduction (Grosse-Kunstleve et al., 2004) to constrain the distribution to the range $p_0(\alpha, \beta, \gamma) = \mathcal{U}(60, 120)$.

**Conditional flows.** Following Chen & Lipman (2024), we train the model to match the conditional flow defined along the geodesic path of the Riemannian manifold:

$$q^{(t)} = \exp_{q^{(0)}}(t \log_{q^{(0)}}(q^{(1)})), \quad \tau^{(t)} = (1-t)\tau^{(0)} + t\tau^{(1)}, \quad \boldsymbol{\ell}^{(t)} = (1-t)\boldsymbol{\ell}^{(0)} + t\boldsymbol{\ell}^{(1)}. \tag{5}$$

Here, $\exp_q$ is the exponential map and $\log_q$ is the logarithmic map at point $q$. From this definition, the conditional vector fields are derived from the time derivatives:

$$u_t(q^{(t)}|q^{(1)}) = \frac{\log_{q^{(t)}}(q^{(1)})}{1-t}, \quad u_t(\tau^{(t)}|\tau^{(1)}) = \frac{\tau^{(1)} - \tau^{(t)}}{1-t}, \quad u_t(\boldsymbol{\ell}^{(t)}|\boldsymbol{\ell}^{(1)}) = \frac{\boldsymbol{\ell}^{(1)} - \boldsymbol{\ell}^{(t)}}{1-t}. \tag{6}$$

**Training objective.** Instead of directly modeling the vector fields, we leverage a closed-form expression that enables re-parameterization of the network to predict the clean data $\boldsymbol{q}_1, \boldsymbol{\tau}_1, \boldsymbol{\ell}_1$ from the intermediate MOF structure $\mathcal{S}^{(t)} = (\boldsymbol{q}_t, \boldsymbol{\tau}_t, \boldsymbol{\ell}_t, \mathcal{B})$. To achieve this, we train a neural network to approximate the clean data, expressed as $(\hat{\boldsymbol{q}}_1, \hat{\boldsymbol{\tau}}_1, \hat{\boldsymbol{\ell}}_1) = \mathcal{F}(\mathcal{S}^{(t)}; \theta)$ with regression on clean data:

$$\mathcal{L}(\theta) = \mathbb{E}_{\mathcal{S}^{(1)} \sim \mathcal{D}} \mathbb{E}_{t \sim \mathcal{U}(0,1)} \Big[ \frac{1}{(1-t)^2} \Big( \lambda_1 \|\log_{\mathbf{q}^{(t)}}(\hat{\boldsymbol{q}}_1) - \log_{\mathbf{q}^{(t)}}(\boldsymbol{q}_1)\|_{SO(3)}^2$$

$$+ \lambda_2 \|\hat{\boldsymbol{\tau}}_1 - \boldsymbol{\tau}_1\|_{\mathbb{R}^3}^2 + \lambda_3 \|\hat{\boldsymbol{\ell}}_1 - \boldsymbol{\ell}_1\|_{\mathbb{R}^3}^2 \Big) \Big], \tag{7}$$

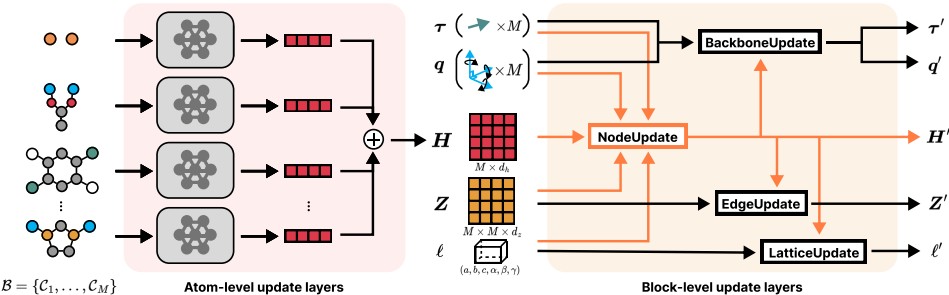

Figure 3: **Overview of our neural network architecture.** Our architecture follows a hierarchical structure, starting with atom-level update layers that encode building block representations into atomic-resolution embeddings. These are followed by block-level update layers, which iteratively refine the roto-translations $(\boldsymbol{q}, \boldsymbol{\tau})$, block features $\boldsymbol{H}$, pairwise features $\boldsymbol{Z}$, and lattice parameters $\boldsymbol{\ell}$. The final output is a prediction of the clean data $(\hat{\boldsymbol{q}}_1, \hat{\boldsymbol{\tau}}_1, \hat{\boldsymbol{\ell}}_1)$.

where $\mathcal{D}$ is the dataset, $\mathcal{U}(0, 1)$ is the uniform distribution defined on an interval $[0, 1]$, and $\lambda_1, \lambda_2, \lambda_3$ are loss coefficients (see Appendix C).

## 4.3 MODEL ARCHITECTURE

Here, we describe the architecture of our neural network $\mathcal{F}(\mathcal{S}^{(t)}; \theta)$ used to predict clean data $(\boldsymbol{q}_1, \boldsymbol{\tau}_1, \ell_1)$. We have two key modules: (1) the atom-level update layers to obtain the building block embeddings from atomic resolution, and (2) the building block-level update layers that aggregate and update information over MOF on building block resolution and predict $\boldsymbol{q}_1$, $\boldsymbol{\tau}_1$, and $\ell_1$ from the final building block embeddings. In what follows, we describe each module one-by-one.

**Atom-level update layers.** The atom-level update layers (Figure 3) process the building block representation $\mathcal{C}_m = (\boldsymbol{a}_m, \boldsymbol{Y}_m)$ for $m = 1, \ldots, M$ to output the building block embedding $h_m$. They are graph neural networks operating on an undirected graph $\mathcal{G}_m = (\mathcal{V}_m, \mathcal{E}_m)$ constructed from adding an edge between a pair of atoms within the cutoff distance of 5Å. It initializes the atom-wise features $\{v_k : k = 1, \ldots, N_m\}$ from atom types and the edge features $\{e_{k,k'} : \{k, k'\} \in \mathcal{E}\}$ from atomic distances. Each layer updates the atom features $\{v_k : k = 1, \ldots, N_m\}$ as follows:

$$v'_k = v_k + \phi_v\left(v_k, \sum_{k' \in \mathcal{N}(k)} \phi_e\left(v_k, v_{k'}, e_{k,k'}\right)\right), \tag{8}$$

where $\{v'_k : k = 1, \ldots, N_m\}$ is the set of updated atom features, $\mathcal{N}(k)$ denotes the neighbor of atom $k$ in the graph $\mathcal{G}_m$, and $\phi_v, \phi_e$ are multi-layer perceptrons (MLPs). Finally, the building block embedding $h_m$ is obtained from applying mean pooling of the node embeddings at the last layer followed by concatenation with sinusoidal time embedding of $t$ (Vaswani, 2017) and an MLP.

**Block-level update layers.** Each layer of the update module (Figure 3) iteratively updates its prediction of $(\boldsymbol{q}, \boldsymbol{\tau}) \in SE(3)^M$ and $\ell$ along with the block features $\boldsymbol{H} = [h_m]_{m=1}^M \in \mathbb{R}^{M \times d_h}$ and the pairwise features $\boldsymbol{Z} = [z_{mm'}]_{m,m'=1}^M \in \mathbb{R}^{M \times M \times d_z}$. The predictions are initialized by the intermediate flow matching output $\boldsymbol{q}_t, \boldsymbol{\tau}_t, \ell_t$, the node features are initialized from the atom-level update layers, and the edge features are initialized as $z_{mm'} = \phi_z(h_m, h_{m'}, \text{dgram}(\|\tau_m - \tau_{m'}\|_2), \text{dgram}(\|\hat{\tau}_m - \hat{\tau}_{m'}\|_2))$, where dgram computes a distogram binning the pairwise distance into equally spaced intervals between 0Å and 20Å. Finally, the block-level update module is defined as follows: $\boldsymbol{H}' = \text{NodeUpdate}(\boldsymbol{H}, \boldsymbol{Z}, \boldsymbol{q}, \boldsymbol{\tau}, \ell)$, $\boldsymbol{Z}' = \text{EdgeUpdate}(\boldsymbol{Z}, \boldsymbol{H}')$, $(\boldsymbol{q}', \boldsymbol{\tau}') = \text{BackboneUpdate}(\boldsymbol{q}, \boldsymbol{\tau}, \boldsymbol{H}')$, $\ell' = \text{LatticeUpdate}(\ell, \boldsymbol{H}')$, where $\boldsymbol{q}', \boldsymbol{\tau}', \boldsymbol{H}', \boldsymbol{Z}', \ell'$ are the updated predictions and the features. Importantly, NodeUpdate operator consists of the newly designed MOFAttention module followed by pre-layer normalization Transformers (Xiong et al., 2020) and MLP with residual connections in between. The EdgeUpdate, BackboneUpdate modules are implemented as in Yim et al. (2023b) and LatticeUpdate is identity function for the lattice parameter except for the last layer. At the final layer, the lattice parameter is predicted from mean pooling of the block features followed by an MLP. Complete details of each update module are in Appendix F.

---

**Algorithm 1** MOFAttention module

---

**Input:** Node features $\boldsymbol{H} = [h_m]_{m=1}^M$, edge features $\boldsymbol{Z} = [z_{mm'}]_{m,m'=1}^M$, rotations $\boldsymbol{q} = [q_m]_{m=1}^M$, translations $\boldsymbol{\tau} = [\tau_m]_{m=1}^M$, lattice parameter $\ell$, number of building blocks $M$, number of heads $N_h$, number of non-rotating channels $N_c$, and number of rotating channels $N_r$.
**Output:** Updated node features $\boldsymbol{H}' = [h'_m]_{m=1}^M$.

1: **for** $h \in \{1, \ldots, N_h\}$ **do**
2:     **for** $m \in \{1, \ldots, M\}$ **do**                                                    ▷ Query, key, values.
3:         $q_{mh}, k_{mh}, v_{mh} = \text{Linear}_h(h_m)$.                                    ▷ $q_{mh}, k_{mh}, v_{mh} \in \mathbb{R}^{N_c}$.
4:         $\tilde{q}_{mhp}, \tilde{k}_{mhp}, \tilde{v}_{mhp} = (q_m, \tau_m) \cdot \text{Linear}_h(h_m)$.          ▷ $\tilde{q}_{mh}, \tilde{k}_{mh}, \tilde{v}_{mh} \in \mathbb{R}^{3N_r}$.
5:     **end for**
6:     $l_h = \text{Linear}_h(\ell)$.                                                        ▷ Lattice structure encoding $l_h \in \mathbb{R}$.
7:     $b_{mm'h} = \text{Linear}_h(z_{mm'})$ for $m, m' \in \{1, \ldots, M\}$.          ▷ Attention bias $b_{mm'h} \in \mathbb{R}$.
8:     $\gamma_h = \text{SoftPlus}(\text{Linear}_h(1))$.                                  ▷ Learnable coefficient $\gamma_h \in \mathbb{R}$
9:     $C_1 = \frac{1}{\sqrt{N_c}}$ and $C_2 = \sqrt{\frac{2}{9N_r}}$.                        ▷ Coefficients $C_1, C_2 \in \mathbb{R}$.
10:    **for** $m, m' \in \{1, \ldots, M\}$ **do**                                      ▷ Attention $a_{mm'h} \in \mathbb{R}$.
11:        $a_{mm'h} = \text{SoftMax}\left(\frac{1}{2}\left(C_1 q_{mh}^\top k_{m'h} + b_{mm'h} + l_h - \frac{C_2\gamma_h}{2}\|\tilde{q}_{mhp} - \tilde{k}_{m'hp}\|^2\right)\right)$.
12:    **end for**
13:    $o_{mh} = \sum_{m'} a_{mm'h} v_{m'h}$ and $\tilde{o}_{mh} = \sum_{m'} a_{mm'h} \tilde{v}_{m'h}$ for $m \in \{1, \ldots, M\}$.  ▷ Aggregate.
14: **end for**
15: $h'_m = \text{Linear}(\text{Concat}_{h,p}(o_{mh}, \tilde{o}_{mhp}, \ell))$ for $m \in \{1, \ldots, M\}$          ▷ Update the node features.

---

In particular, our MOFAttention module is modification of the invariant point attention module proposed by Jumper et al. (2021) for processing protein frames. Our modification consists of adding the lattice parameter as input and simplification by removing the edge aggregation information. In particular, the lattice parameter is embedded using a linear layer and added as an offset for the attention matrix between the building blocks. We provide more details in Algorithm 1.

## 5 EXPERIMENTS

The goal of our experiments is to address two questions. **Accuracy:** How does the structure prediction accuracy of MOFFLOW compare to other approaches? **Scalability:** How does the performance of MOFFLOW vary with an increasing number of atoms and building blocks?

To address the first question, Section 5.1 compares the structure prediction accuracy of MOFFLOW against both conventional and deep learning-based methods. Furthermore, Section 5.2 evaluates whether MOFFLOW can capture essential MOF properties, further validating the accuracy of its predictions. The second question is answered in Section 5.3, where we analyze the performance of MOFFLOW with increasing system size. Additionally in Section 5.4, we compare MOFFLOW to the self-assembly algorithm (Fu et al., 2023) by integrating it with our approach.

### 5.1 STRUCTURE PREDICTION

**Dataset.** We use the dataset from Boyd et al. (2019), containing 324,426 MOF structures. Following Fu et al. (2023), we apply the `metal-oxo` decomposition of `MOFid` (Bucior et al., 2019) to decompose each structure into building blocks. After filtering structures with fewer than 200 blocks, we split the data into train/valid/test sets in an 8:1:1 ratio. Full data statistics are in Appendix A.

**Baselines.** We compare our model with two types of methods: optimization-based algorithms and deep learning-based methods. For traditional approach, we use CrySPY (Yamashita et al., 2021) to implement the random search (RS) and evolutionary algorithm (EA). For deep learning, we benchmark against DiffCSP (Jiao et al., 2024a), which generates structures based on atom types. MOF-specific methods are excluded due to lack of public availability.

**Metrics.** We evaluate using match rate (MR) and root mean square error (RMSE). We compare the samples to the ground truth using `StructureMatcher` class of `pymatgen` (Ong et al., 2013). Two sets of threshold for stol, ltol, and angle_tol are used: $(0.5, 0.3, 10.0)$ in alignment with the

Table 1: **Structure prediction accuracy.** We compare optimization-based methods (RS, EA), a deep generative model (DiffCSP), and MOFFLOW. Due to computational constraints, RS and EA were tested on 100 and 15 samples, respectively, while DiffCSP and MOFFLOW were evaluated on the full test set (30,880 structures). MR is the match rate and RMSE is the root mean squared error; - indicates no match. stol is the site tolerance for matching criteria. The reported time is the average per structure. MOFFLOW outperforms all baselines in MR, RMSE, and generation time.

| | # of samples | stol = 0.5 | | stol = 1.0 | | Avg. time (s)↓ |
|---|---|---|---|---|---|---|
| | | MR (%) ↑ | RMSE ↓ | MR (%) ↑ | RMSE ↓ | |
| RS (Yamashita et al., 2021) | 20 | 0.00 | - | 0.00 | - | 332 |
| EA (Yamashita et al., 2021) | 20 | 0.00 | - | 0.00 | - | 1959 |
| DiffCSP (Jiao et al., 2024a) | 1 | 0.09 | 0.3961 | 23.12 | 0.8294 | 5.37 |
| | 5 | 0.34 | 0.3848 | 38.94 | 0.7937 | 26.85 |
| MOFFLOW (Ours) | 1 | 31.69 | 0.2820 | 87.46 | 0.5183 | **1.94** |
| | 5 | **44.75** | **0.2694** | **100.0** | **0.4645** | 5.69 |

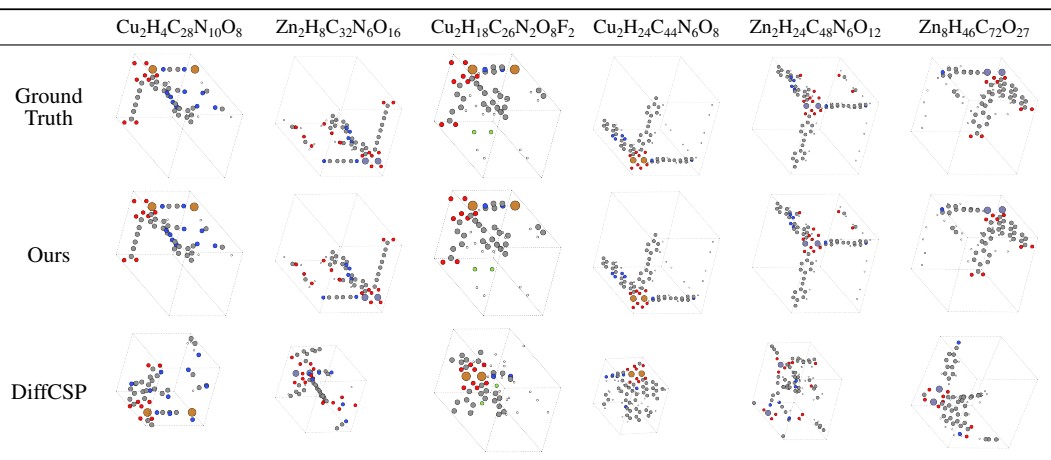

Figure 4: **Visualization of the predicted MOF structures.** We select structures from the 20 candidates with the lowest RMSE. The lattice is scaled to reflect the relative sizes. MOFFLOW accurately generates high-quality predictions with accurate atomic positions and lattice configuration.

general CSP literature (Jiao et al., 2024a;b; Chen & Lipman, 2024) and $(1.0, 0.3, 10.0)$ to account for the difficulty of predicting large structures. MR is the proportion of matched structures and RMSE is the root mean squared displacement normalized by the average free length per atom. We also measure the time required to generate $k$ samples, averaged across all test sets.

**Implementation.** Both RS and EA use CHGnet (Deng et al., 2023) for structure optimization. We generate 20 samples for RS, while EA starts with 5 initial, 20 populations, and 20 generations. Due to the high computational cost for large crystals, generating more samples was not feasible. For DiffCSP, we follow the hyperparameters from Jiao et al. (2024a) and train for 200 epochs. Our method uses an AdamW optimizer (Loshchilov, 2017) with a learning rate of $10^{-4}$ and $\beta = (0.9, 0.98)$. We use a maximum batch size of 160 and run inference with 50 integration steps (see Appendix G for analysis on integration steps). We generate 1 and 5 samples for DiffCSP and our method. Implementation details are in Appendix C.

**Results.** Table 1 presents the results, where MOFFLOW outperforms all baselines. Optimization-based methods yield zero MR, highlighting the challenge of using the conventional atom-based approach for large systems. DiffCSP also performs poorly, underscoring the need to incorporate building block information in MOF structure prediction. While we achieve a 100% MR at stol = 1.0, this threshold is too lenient for practical application; however, we include it for multi-level comparison. Visualizations comparing samples from MOFFLOW and DiffCSP are shown in Figure 4.

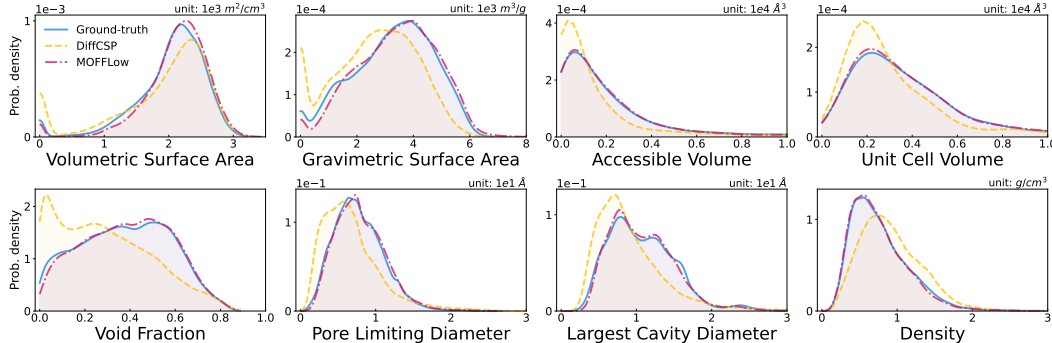

Figure 5: **Property distributions.** We compare the distributions of key MOF properties for ground-truth, MOFFLOW, and DiffCSP. The distributions are histograms smoothed by kernel density estimation. Property units are displayed in the top-right corner of each plot. MOFFLOW (red) closely aligns with the ground-truth distribution (blue), while DiffCSP (yellow) shows noticeable deviations. These results highlight that MOFFLOW's ability to accurately capture essential MOF properties.

## 5.2 PROPERTY EVALUATION

In this section, we demonstrate that MOFFLOW accurately captures the key properties of ground-truth MOF structures, offering a more detailed assessment of prediction quality beyond match rate and RMSE. These properties are crucial for various MOF applications, such as gas storage and catalysis. Specifically, we evaluate volumetric surface area (VSA), gravimetric surface area (GSA), largest cavity diameter (LCD), pore limiting diameter (PLD), void fraction (VF), density (DST), accessible volume (AV), and unit cell volume (UCV). Definitions and the details of each property are provided in Appendix B.

We compare our model to DiffCSP as a representative general CSP approach. We exclude optimization-based baselines as they did not yield meaningful results. For each test structure in Section 5.1, we generate a single sample with 50 integration steps, then evaluate the properties of predicted and ground-truth structures using Zeo++ (Willems et al., 2012). We evaluate the models with RMSE and distributional differences. We do not filter our samples with the match criteria from Section 5.1.

**Results.** Table 2 shows that MOFFLOW consistently yields lower errors than DiffCSP across all evaluated properties. This demonstrates its ability to produce high-quality predictions while preserving essential MOF characteristics. Additionally, Figure 5 visualizes the property distributions, where our model closely reproduces the ground-truth distributions and captures key characteristics. In contrast, DiffCSP frequently reduces volumetric surface area and void fraction to zero, highlighting the limitations of conventional approaches in accurately modeling MOF properties.

Table 2: **Property evaluation.** Average RMSE computed between the ground-truth and generated structures for MOF-FLOW and DiffCSP. MOFFLOW achieve lower error across all properties, demonstrating its ability to generate high-quality samples that accurately capture MOF properties.

|  | RMSE $\downarrow$ | |
| --- | --- | --- |
|  | MOFFLOW | DiffCSP |
| VSA $(m^2/cm^3)$ | **264.5** | 796.9 |
| GSA $(m^2/g)$ | **331.6** | 1561.9 |
| AV $(\mathring{A}^3)$ | **530.5** | 3010.2 |
| UCV $(\mathring{A}^3)$ | **569.5** | 3183.4 |
| VF | **0.0285** | 0.2167 |
| PLD $(\mathring{A})$ | **1.0616** | 4.0581 |
| LCD $(\mathring{A})$ | **1.1083** | 4.5180 |
| DST $(g/cm^3)$ | **0.0442** | 0.3711 |

## 5.3 SCALABILITY EVALUATION

Here, we demonstrate that MOFFLOW enables structure prediction for large systems, which is a challenge for general CSP methods. To evaluate how performance scales with system size, we analyze match rates as a function of the number of atoms and building blocks. We compare our results with DiffCSP as the representative of the general CSP approach. We generate single samples for each test structure and use thresholds $(1.0, 0.3, 10.0)$ for visibility.

**Results.** Figure 6 presents our findings, with the $x$-axis representing the number of atoms, binned in ranges of $(t, t + 200]$. The final bin includes all atom counts beyond the last range, without an upper limit. MOFFlow consistently outperforms DiffCSP across all atom ranges. While our approach shows only gradual performance degradation as atom count increases, DiffCSP suffers a sharp decline for systems with more than 100 atoms and fails to predict structures with over 200 atoms. In contrast, our method maintains a high match rate even for structures exceeding 1,000 atoms per unit cell, highlighting the effectiveness of leveraging building block information for MOF structure prediction. Additionally, Figure 6 shows how our match rate scales with the number of building blocks. The results show minimal performance degradation, demonstrating that our model effectively handles larger numbers of building blocks and efficiently scales to large crystal structures.

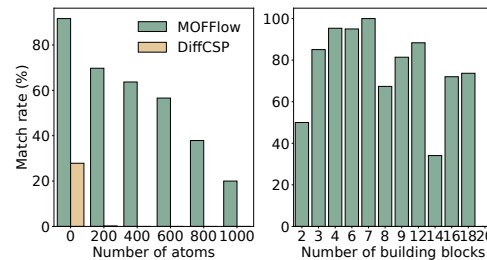

Figure 6: **Scalability evaluation.** *(left)* Match rate comparison between MOFFLOW and DiffCSP by atom count. MOFFLOW preserves high match rates across all bins, while DiffCSP drops sharply beyond 200 atoms. *(right)* Match rate of MOFFLOW by building block count, with our method performing well even for complex structures with many blocks. These results highlight the scalability of our approach.

## 5.4 COMPARISON TO SELF-ASSEMBLY ALGORITHM

To make our evaluation more comprehensive, we also consider the self-assembly algorithm used by Fu et al. (2023) as a baseline, although the performance is not directly comparable. The self-assembly (SA) algorithm is an optimization-based method that predicts the rotation $q$ by maximizing the overlap between building block connection points. Since the algorithm requires $\tau$, $\ell$, and $\mathcal{C}$ as input, it is not directly applicable for structure prediction on its own. Therefore, we conduct an ablation by combining the self-assembly algorithm with our predicted values of $\tau$ and $\ell$. We note that the self-assembly algorithm defines the centroid as the center of mass of the connection points, and we account for this offset in our implementation.

Table 3: **Comparison with self-assembly (SA) algorithm.** Since SA can only predict rotations, we provide translations and lattice predicted by MOFFLOW for fair comparison. MOFFLOW alone achieves higher accuracy and faster inference times than SA.

| | MR (%) ↑ | RMSE ↓ | Time (s) ↓ |
|---|---|---|---|
| SA | 30.04 | 0.3084 | 4.75 |
| Ours | **31.69** | **0.2820** | **1.94** |

**Results.** Table 3 shows that MOFFlow alone outperforms the combination with the self-assembly algorithm, indicating that learning the building block orientations leads to more accurate MOF structure predictions than heuristic-based overlap optimization. Additionally, our method offers faster inference, further demonstrating its efficiency compared to optimization-based approaches. A comprehensive comparison is provided in Appendix H.

## 6 DISCUSSIONS

We propose MOFFLOW, a building block-based approach for predicting the structure of metal-organic frameworks (MOFs). Our approach significantly outperforms general crystal structure prediction algorithms – in both quality and efficiency – that fail to account for the modularity of MOFs. Additionally, MOFFLOW is scalable, successfully predicting structures composed of up to thousands of atoms.

**Limitations.** MOFFLOW was evaluated only on the hypothetical database (Boyd et al., 2019), highlighting the need for evaluation on more challenging real-world datasets. Additionally, we assume that the local building block structures are known (i.e., the rigid body assumption), which may be impractical for rare building blocks whose structural information is missing from existing libraries (e.g., Gibaldi et al. (2022)) or is inaccurate. Finally, MOFFLOW is not invariant to periodic transformations of the input; explicitly modeling periodic invariance could further improve its performance.

REPRODUCIBILITY

We describe experimental details and hyperparameters in Appendix C. We provide our codes and model checkpoint in `https://github.com/nayoung10/mofflow`.

ACKNOWLEDGMENTS

This work was partly supported by Institute for Information & communications Technology Technology Planning & Evaluation(IITP) grant funded by the Korea government(MSIT) (RS-2019-II190075, Artificial Intelligence Graduate School Support Program(KAIST)), National Research Foundation of Korea(NRF) grant funded by the Ministry of Science and ICT(MSIT) (No. RS-2022-NR072184), and GRDC(Global Research Development Center) Cooperative Hub Program through the National Research Foundation of Korea(NRF) grant funded by the Ministry of Science and ICT(MSIT) (No. RS-2024-00436165).

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

# A  DATA STATISTICS

In this section, we present the data statistics to represent the characteristics of the MOF dataset. We consider the MOF dataset from Boyd et al. (2019) and the dataset is generated by the MOF-generating algorithms based on the topology from graph theory. The dataset is distributed on the MATERIALSCLOUD. As mention in Section 5.1, we uses filtered strucutures with fewer than 200 blocks. The dataset is divided into train, valid and test in an 8:1:1 ratio. The statistic of data splits are represented in the Tables 4, 5 and 6.

| **Property** (number of samples $= 247,066$) | **Min** | **Mean** | **Max** |
|---|---|---|---|
| number of species / atoms | 3 / 20 | 5.3 / 151.5 | 8 / 2208 |
| working capacity (vacuum) $[10^{-3}\text{mol/g}]$ | -0.2618 | 0.4177 | 4.8355 |
| working capacity (temperature) $[10^{-3}\text{mol/g}]$ | -0.2712 | 0.2843 | 4.4044 |
| volume $[\text{Å}^3]$ | 534.5 | 5496.7 | 193341.7 |
| density $[\text{atoms}/\text{Å}^3]$ | 0.0737 | 0.7746 | 4.0966 |
| lattice $a, b, c$ $[\text{Å}]$ | 6.13 / 8.27 / 8.56 | 13.81 / 16.47 / 20.39 | 57.83 / 57.80 / 61.62 |
| lattice $\alpha, \beta, \gamma$ $[°]$ | 59.76 / 59.99 / 59.97 | 91.08 / 91.05 / 90.75 | 120.29 / 120.01 / 120.02 |

Table 4: The statistics of the train split of MOF dataset.

| **Property** (number of samples $= 30,883$) | **Min** | **Mean** | **Max** |
|---|---|---|---|
| number of species / atoms | 3 / 16 | 5.3 / 152.4 | 8 / 2256 |
| working capacity (vacuum) $[10^{-3}\text{mol/g}]$ | -0.2510 | 0.4163 | 5.1152 |
| working capacity (temperature) $[10^{-3}\text{mol/g}]$ | -0.1210 | 0.2834 | 4.4589 |
| volume $[\text{Å}^3]$ | 534.5 | 5538.1 | 118597.4 |
| density $[\text{atoms}/\text{Å}^3]$ | 0.11 | 0.77 | 4.33 |
| lattice $a, b, c$ $[\text{Å}]$ | 6.86 / 8.43 / 8.57 | 13.85 / 16.53 / 20.39 | 48.29 / 55.11 / 60.97 |
| lattice $\alpha, \beta, \gamma$ $[°]$ | 59.98 / 59.99 / 59.99 | 91.08 / 91.00 / 90.73 | 120.11 / 120.01 / 120.02 |

Table 5: The statistics of the valid split of MOF dataset.

| **Property** (number of samples $= 30,880$) | **Min** | **Mean** | **Max** |
|---|---|---|---|
| number of species / atoms | 3 / 22 | 5.3 / 149.3 | 8 / 2368 |
| working capacity (vacuum) $[10^{-3}\text{mol/g}]$ | -0.1999 | 0.4193 | 4.6545 |
| working capacity (temperature) $[10^{-3}\text{mol/g}]$ | -0.1318 | 0.2858 | 3.9931 |
| volume $[\text{Å}^3]$ | 536.4 | 5401.1 | 124062.6 |
| density $[\text{atoms}/\text{Å}^3]$ | 0.108 | 0.777 | 4.074 |
| lattice $a, b, c$ $[\text{Å}]$ | 6.86 / 8.34 / 8.56 | 13.74 / 16.39 / 20.26 | 49.86 / 49.88 / 60.95 |
| lattice $\alpha, \beta, \gamma$ $[°]$ | 59.91 / 60.00 / 59.99 | 91.00 / 90.98 / 90.76 | 120.16 / 120.01 / 120.01 |

Table 6: The statistics of the test split of MOF dataset.

# B GLOSSARY

**Structural properties.** In this section, we introduce the structural properties we measured. These properties were calculated using the Zeo++ software package developed by Willems et al. (2012). Zeo++ provides high-throughput geometry-based analysis of crystalline porous materials, calculating critical features such as pore diameters, surface area, and accessible volume, all of which are essential for evaluating material performance in applications such as gas storage and catalysis.

Specifically, we calculated properties including volumetric surface area (VSA), the surface area per unit volume; gravimetric surface area (GSA), which represents the surface area per unit mass; the largest cavity diameter (LCD), which represents the diameter of the largest spherical cavity within the material; the pore limiting diameter (PLD), defined as the smallest passage through which molecules must pass to access internal voids; the void fraction (VF) (Martin & Haranczyk, 2014), which is the ratio of total pore volume in the structure to the total cell volume; the density (DST), which refers to the mass per unit volume of the material; the accessible volume (AV), indicating the volume available to the center of a given probe molecule within the pores; and the unit cell volume (UCV), representing the total volume of the repeating unit cell in the crystal structure. These parameters provide critical insights into the MOF's porosity, surface area, and ability to store and transport gases, and recent study shows the correlation with these properties with the bulk material (Krishnapriyan et al., 2020).

# C IMPLEMENTATION DETAILS

**Training details.** We use the TimestepBatch algorithm (Yim et al., 2023b) to simplify batch construction. This method generates a batch by applying multiple noise levels $t \in [0, 1]$ to a single data instance, ensuring uniform batch size. To manage memory constraints, we cap the batch size with $N^2$, where $N$ is the number of atoms.

**Hyperparameters.** Table 7 and Table 8 shows model and training hyperparameters for MOFFLOW, respectively. In practice, we generate $q$, $k$, and $v$ from $h$ independently, allowing each to have a distinct dimension. The non-rotating channels are represented as a tuple, with $qk$ and $v$ specified in that order. We set the log-normal distribution parameters for lattice lengths to $\mu = (2.55, 2.75, 2.96)$ and $\sigma = (0.3739, 0.3011, 0.3126)$, computed from the training data with the closed-form maximum likelihood estimation.

**Baselines.** Here, we provide the implementation details of the baselines. Unless specified otherwise, all hyperparameters follow their default settings.

- **DiffCSP** (Jiao et al., 2024a): To address memory constraints, we replaced fully connected edge construction with a radius graph (cutoff: 5Å) and used a batch size of 8. The model was trained on a 24GB NVIDIA RTX 3090 GPU for 5 days until convergence.

- **RS & EA**: Both random search (RS) and the evolutionary algorithm (EA) were implemented with CRYSPY (Yamashita et al., 2021) and energy-based optimization with CHGNet (Deng et al., 2023). For RS, we generated 20 structures per sample with a symmetry-based search. EA began with 5 initial RS runs and performed up to 20 generations with a population size of 20, 10 crossovers, 4 permutations, 2 strains, and 2 elites. A tournament selection function with a size of 4 was employed.

FlowMM (Miller et al., 2024) and EquiCSP (Lin et al., 2024) were excluded from our baselines due to their high resource demands when adapted for MOF structure prediction, requiring more than 20 days of training on an 80GB A100 GPU.

**Computational resources.** Table 9 summarizes the computational resources required to train learning-based models. Notably, the TimestepBatch implementation of MOFFLOW requires longer training times than DiffCSP in terms of GPU hours. To address this inefficiency, we also release a refactored Batch version of MOFFLOW with details in Appendix D.

**Codebase.** Our implementation is built on https://github.com/gcorso/DiffDock, https://github.com/vgsatorras/egnn, https://github.com/microsoft/MOFDiff, and https://github.com/microsoft/protein-frame-flow. We appre-

ciate the authors (Yim et al., 2023a;b; 2024; Fu et al., 2023; Corso et al., 2023; Satorras et al., 2021) for their contributions.

Table 7: Model hyperparameters of MOFFLOW

| Hyperparameter | Value |
|---|---|
| Atom-level node dimension | 64 |
| Atom-level edge dimension | 64 |
| Atom-level cutoff radius | 5 |
| Atom-level maximum atoms | 100 |
| Atom-level update layers | 4 |
| Block-level node dimension | 256 |
| Block-level edge dimension | 128 |
| Block-level time dimension | 128 |
| Block-level update layers | 6 |
| MOFAttention number of heads | 24 |
| MOFAttention rotating channels | 256 |
| MOFAttention non-rotating channels | (8, 12) |

Table 8: Training hyperparameters of MOFFLOW

| Hyperparameter | Value |
|---|---|
| loss coefficient $\lambda_1$ ($q$) | 1.0 |
| loss coefficient $\lambda_2$ ($\tau$) | 2.0 |
| loss coefficient $\lambda_3$ ($\ell$) | 0.1 |
| batch size | 160 |
| maximum $N^2$ | 1,600,000 |
| optimizer | AdamW |
| initial learning rate | 0.0001 |
| betas | (0.9, 0.98) |
| learning rate scheduler | ReduceLROnPlateau |
| learning rate patience | 30 epochs |
| learning rate factor | 0.6 |

Table 9: **Computational resources.** Comparison of batch size, GPU type ($\times$ number), GPU memory utilization (GB), and training time (d: days, h: hours) for training learning-based models.

| | Batch size | GPU Type | GPU memory (GB) | Training time |
|---|---|---|---|---|
| DiffCSP | 8 | 24GB 3090 ($\times$1) | 23.24 - 23.96 | 5d 2h |
| MOFFLOW (TimestepBatch) | 160 | 24GB 3090 ($\times$8) | 6.67 - 23.95 | 5d 15h |
| MOFFLOW (Batch) | 160 | 24GB 3090 ($\times$8) | 21.62 - 23.80 | 1d 17h |

# D    BATCH IMPLEMENTATION OF MOFFLOW

While the TimestepBatch algorithm (Yim et al., 2023b) simplifies implementation, it slows convergence due to its effective batch size of 1 (i.e., each batch contains noise-perturbed variants of a single instance). To address this limitation, we introduce the Batch implementation, which processes multiple data instances per batch, aligning with standard practice. As shown in Table 10, Batch achieves comparable performance to TimestepBatch while significantly reducing training time from $1087.25$ to $332.74$ GPU hours. It also reduces inference time from $1.94$ to $0.1932$ seconds.

Table 10: **Comparison of** TimestepBatch **and** Batch **implementations of MOFFLOW.** The Batch implementation achieves comparable performance while significantly reducing training time (GPU hours) and inference time (seconds). Inference time is reported as the average per test instance, calculated by dividing the total elapsed time by the size of the test set.

|  | Training time (h) | Inference time (s) | Match rate (%) | RMSE |
|---|---|---|---|---|
| TimestepBatch | 1087.25 | 1.94 | 31.69 | 0.2820 |
| Batch | 332.74 | 0.19 | 32.73 | 0.2743 |

# E    DEFINING LOCAL COORDINATES OF BUILDING BLOCKS

Following, Gao & Günnemann (2022), we use principle component analysis (PCA) as our backbone since it is $SO(3)$-equivariant up to a sign. Specifically, if we denote $\text{PCA}(X) = [e_1, e_2, e_3]$, in the order of decreasing eigenvalues, $\forall Q \in SO(3)$,

$$\text{PCA}(\boldsymbol{X}Q^\top) = c \odot Q\,\text{PCA}(\boldsymbol{X}), \quad c \in \{-1, +1\}^3$$

that is, the sign is not preserved upon rotation. To define a consistent direction, Gao & Günnemann (2022) suggests the use of an equivariant vector function $v(\boldsymbol{a}, \boldsymbol{X})$ as

$$\tilde{e}_i = \begin{cases} e_i, & \text{if } v(\boldsymbol{a}, \boldsymbol{X})^\top e_i \geq 0 \\ -e_i, & \text{otherwise.} \end{cases} \tag{9}$$

Then, the final equivariant axes is defined as $\mathcal{R} = [\tilde{e}_1, \tilde{e}_2, \tilde{e}_3]$ where $\tilde{e}_3 = \tilde{e}_1 \times \tilde{e}_2$.

However, we find that this definition is insufficient for our application where some building blocks are 2-dimensional exhibit symmetry with respect to the origin and thus have $v(\boldsymbol{a}, \boldsymbol{X}) = 0$. For such cases, we define

$$v_{\text{sym}}(\boldsymbol{X}) = \underset{x \in \boldsymbol{X}}{\arg\min}\{\|x\|^2 | x \neq 0\}$$

– i.e., the vector from the centroid to the closest atom. Since the building blocks are symmetric, $f$ still fulfills Equation (3) up to permutation, which is handled by GNN and Transformers.

# F  MODEL ARCHITECTURE

Here, we provide the details of the NodeUpdate, EdgeUpdate, and BackboneUpdate modules. Our implementation follows Yim et al. (2023b); Jumper et al. (2021), with the exception of the MOFAttention module and the Transformers, where we use a pre-layer normalized version (Xiong et al., 2020). Each module is introduced with our notation. The function $R(a, b, c, d)$ in Algorithm 4 is defined as

$$R(a,b,c,d) = \begin{pmatrix} (a^n)^2+(b^n)^2-(c^n)^2-(d^n)^2 & 2b^n c^n-2a^n d^n & 2b^n d^n+2a^n c^n \\ 2b^n c^n+2a^n d^n & (a^n)^2-(b^n)^2+(c^n)^2-(d^n)^2 & 2c^n d^n-2a^n b^n \\ 2b^n d^n-2a^n c^n & 2c^n d^n-2a^n b^n & (a^n)^2-(b^n)^2-(c^n)^2+(d^n)^2 \end{pmatrix}. \tag{10}$$

---

**Algorithm 2** NodeUpdate Module

---

**Input:** $(q, \tau, H, Z, \ell)$
**Output:** $H'$

1: $\tilde{H} \leftarrow \text{LayerNorm}(\text{MOFAttention}(q, \tau, H, Z, \ell) + H)$
2: $\tilde{H} \leftarrow \text{Concat}(\tilde{H}, \text{Linear}(H^{(0)}))$
3: $\tilde{H} \leftarrow \text{Linear}(\text{Transformers}(\tilde{H})) + H^{(\ell)}$
4: $H' \leftarrow \text{MLP}(\tilde{H})$

---

**Algorithm 3** EdgeUpdate Module

---

**Input:** $(Z, H')$
**Output:** $Z'$

1: **for** $m = 1, \ldots, M$ **do**
2:      $\tilde{h}_m \leftarrow \text{Linear}(H^{(\ell+1)})$
3:      **for** $m' = 1, \ldots M$ **do**
4:          $\tilde{z}_{mm'} \leftarrow \text{Concat}(\tilde{h}_m, \tilde{h}_{m'}, z_{mm'})$
5:      **end for**
6:      $\tilde{Z} \leftarrow [\tilde{z}_{mm'}]_{m,m'=1}^{M}$
7:      $Z' \leftarrow \text{LayerNorm}(\text{MLP}(\tilde{Z}))$
8: **end for**

---

**Algorithm 4** BackboneUpdate Module

---

**Input:** $(q, \tau, H')$
**Output:** $q', \tau'$

1: **for** $m = 1, \ldots, M$ **do**
2:      $(b, c, d, \tilde{\tau}_m) \leftarrow \text{Linear}(h_m)$
3:      $(a, b, c, d) \leftarrow (1, b, c, d)/\sqrt{1+b+c+d}$
4:      $\tilde{q}_m \leftarrow R(a, b, c, d)$
5:      $(q'_m, \tau'_m) \leftarrow (q_m, \tau_m) \cdot (\tilde{q}, \tilde{\tau})$
6: **end for**

---

## G   EFFECT OF INTEGRATION STEPS

Here, we investigate how the number of sampling integration steps affects MOFFLOW's performance. We randomly select 1,000 structures from the test set used in Section 5.1 and evaluate the match rate, RMSE, and average sampling time for varying integration steps: $(5, 10, 50, 100, 200)$. For each structure, We generate a single sample and set the thresholds for stol, ltol, and angle_tol to $(0.5, 0.3, 10.0)$.

**Results.** Figure 7 presents our results. Notably, the performance peaks around 10 and 50 integration steps, with a slight decline observed for higher step counts. This aligns with the trends reported by Miller et al. (2024). Based on these results, we use 50 integration steps in our main experiments, which yield the highest match rate of 31.1%, a low RMSE of 0.2821, and a fast sampling time of 1.267 seconds.

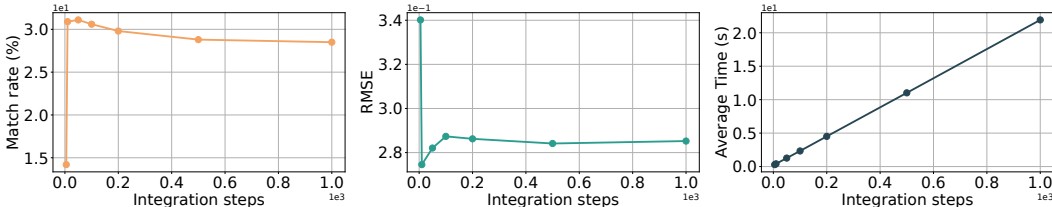

Figure 7: **Effect of integration steps on match rate, RMSE, and average sampling time.** Performance is highest at 10 and 50 integration steps. We select 50 steps for the main experiments due to its optimal balance of the highest match rate, low RMSE, and efficient sampling time.

## H   COMPARISON WITH SELF-ASSEMBLY ALGORITHM

We compare MOFFLOW's scalability and sampling efficiency with the self-assembly algorithm (Fu et al., 2023), highlighting their respective strengths and limitations.

**Scalability.** Since both methods operate at the building block level, we compare their match rates as a function of the number of building blocks. The experimental settings follow Section 5.4. Figure 8a shows that while MOFFLOW achieves a higher overall match rate (31.69% vs. 27.14%), the self-assembly algorithm scales better for structures with more building blocks.

**Sampling efficiency.** Figure 8b compares the assembly times of the two methods. MOFFLOW (*left*) demonstrates significantly faster sampling, below 8 seconds per sample. In contrast, the self-assembly algorithm (*right*) often requires over 2000 seconds per sample. Additionally, the self-assembly algorithm's performance varies widely with initialization, with average assembly times ranging from 4.75 to 14.62 seconds per trial, reflecting its sensitivity to initial conditions.

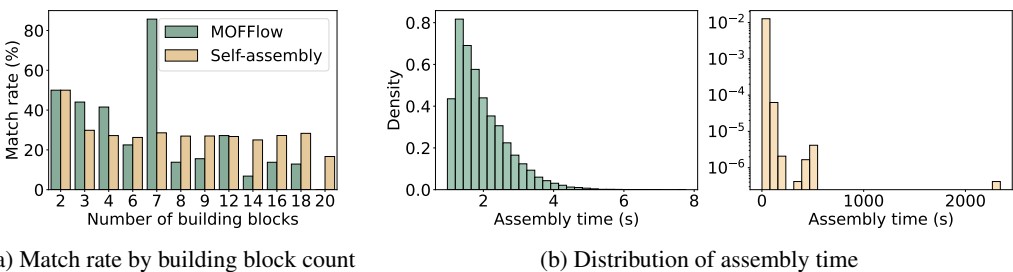

(a) Match rate by building block count        (b) Distribution of assembly time

Figure 8: **Comparison of scalability and efficiency between MOFFLOW and the self-assembly algorithm.** (a) Match rate across varying building block counts. MOFFLOW achieves higher match rates overall, but the self-assembly algorithm performs better for structures with a large number of building blocks. (b) Assembly time distributions for *(left)* MOFFLOW and *(right)* the self-assembly algorithm, highlighting the significantly faster inference speed of MOFFLOW.

