# OpenReview forum: "MOFFlow: Flow Matching for Structure Prediction of Metal-Organic Frameworks"
_ICLR.cc/2025/Conference — ICLR 2025 Poster_

### Official Review · Reviewer_1rvy · 2024-11-02

**Soundness:** 3
**Presentation:** 3
**Contribution:** 3
**Rating:** 6
**Confidence:** 5

**Summary:**

This paper proposes MOFFlow, a flow matching framework for the structure prediction of Metal Organic Frameworks (MOFs). The approach decomposes MOFs into blocks and jointly generates the entire lattice, along with the rotation and translation of each block, using flow matching on different Riemann manifolds. The results demonstrate the superiority of the proposed hierarchical flow matching framework over previous baselines.

**Strengths:**

1. The paper addresses the relatively novel problem of MOF generation and is the first to design an end-to-end framework for jointly generating both the lattice and block-level translations and rotations.
2. The authors construct benchmarks with easily calculable metrics for evaluation, and the results clearly showcase the model's effectiveness.

**Weaknesses:**

1. The discussion on periodicity in MOFs is insufficient. Periodicity is a crucial feature in crystallography. Apart from explicitly modelling the lattice parameters, previous works also adopt multi-edge connections [1] or the fractional coordinate system [2] to address this problem. Additional details on modeling periodicity would enhance the paper. For example:
- In Line 235-237, the authors mention the mean-free system to maintain translation invariance. However, constructing mean-free or center-of-mass (CoM) free systems for periodic boundary conditions (PBCs) differs from non-periodic systems [3] and warrants further discussion.
- The backbone model is based on a Cartesian coordinate system, which necessitates careful design of edge connections to capture intra-cell interactions.

2. The design of the backbone model lacks clarity. It is unclear whether the atom-level and block-level modules are updated sequentially or alternatively. Including a figure of the overall architecture would be beneficial.

[1] Fu, Xiang, et al. "MOFDiff: Coarse-grained Diffusion for Metal-Organic Framework Design."

[2] Jiao, Rui, et al. "Crystal structure prediction by joint equivariant diffusion."

[3] Lin, Peijia, et al. "Equivariant Diffusion for Crystal Structure Prediction."

**Questions:**

1. How many different types of building blocks can be decomposed from the dataset?
2. By generating only translations and rotations instead of fine-grained structures, the proposed method assume building blocks as rigid bodies. Could this assumption potentially limit the model's performance?

---

> ### Author Response · Authors · 2024-11-23
>
> Dear reviewer 1rvy,
>
> We deeply appreciate your thorough review and constructive comments on our manuscript. In the revised manuscript, we have marked the changes in **blue**. Below, we address the reviewer's concerns in detail.
>
> ---
>
> **W1. The discussion on periodicity in MOFs is insufficient (e.g., periodic CoM, periodic edge connections). Additional details on modeling periodicity would enhance the paper.**
>
> We appreciate the reviewer’s insightful comment. MOFFlow addresses periodicity implicitly through learning similar to [1]. We attempted to directly model periodicity before submission but encountered training instability and poor performance. We elaborate our findings below.
>
> As noted by the reviewer, periodicity on Cartesian coordinate system can be achieved by constructing periodic invariant edges and using model architectures that update periodic invariant features ([3, 4, 5]). However, implementing this approach in our framework revealed two challenges:
> 1. **Training instability:** We use a joint prediction framework, where the lattice parameter (L) is not fixed during periodic edge construction. This contrasts with previous works that either use ground-truth L (Matformer[3]) or generate L in a separate conditional step before constructing periodic edges (Symat[4], MOFDiff[5]). The absence of a fixed L likely caused substantial training instability.
> 2. **Limited expressivity of the model:** Architectures designed for periodic invariance (i.e., those updating invariant edges instead of explicit coordinates) performed poorly, likely due to limited model expressivity. This aligns findings in MCF[6] and AlphaFold3[7], where hard-coded inductive biases can degrade performance or provide no benefits.
>
> These challenges highlight the difficulty of directly modeling periodicity within our framework, though we remain committed to addressing this problem. In particular, we find that the periodic CoM approach [2] mentioned by the reviewer presents a promising direction for future work.
>
> **W2. Design of backbone model lacks clarity. It is unclear whether atom-level and block-level modules are updated sequentially or alternatively. A figure for overall architecture would be beneficial.**
>
> We apologize for the lack of clarity. The update process is sequential: the atom-level module generates building block embeddings followed by iterative updates of the block-level module. In response to your suggestion, we have re-drawn Figure 3 in the revised manuscript to clearly illustrate the overall architecture. We appreciate your feedback, which helped improve the presentation of our model.
>
> ---
>
> **Q1. How many different types of building blocks can be decomposed from the dataset?**
>
> The BW-DB dataset contains approximately 2 million building blocks [5]. This large number arises because building blocks with the same atomic composition $\mathbf{a}_m$ can have slightly different 3D coordinates $\mathbf{Y}_m$ due to variations in torsion angles of rotatable bonds.
>
> **Q2. By generating only translations and rotations instead of fine-grained structures, the proposed method assume building blocks as rigid bodies. Could this assumption potentially limit the model's performance?**
>
> Thank you for the thoughtful question. We believe the rigid-body assumption is sufficient for the structure prediction task, as many commonly used MOF building blocks are known to remain rigid during the assembly process [8].
>
> That said, extending our model to predict the dynamics of MOFs would likely benefit from relaxing this assumption and explicitly modeling torsional angles. This would be an exciting direction for future work.
>
> ---
>
> ### Reference
>
> [1] Köhler et al., Rigid Body Flows for Sampling Molecular Crystal Structures, ICML 2023.
> [2] Lin et al., Equivariant Diffusion for Crystal Structure Prediction., ICML 2024.
> [3] Yan et al., Periodic Graph Transformers for Crystal Material Property Prediction, NeurIPS 2022.
> [4] Luo et al., Towards Symmetry-Aware Generation of Periodic Materials, NeurIPS 2023.
> [5] Fu et al., MOFDiff: Coarse-grained Diffusion for Metal-Organic Framework Design, ICLR 2024.
> [6] Wang et al., Swallowing the Bitter Pill: Simplified Scalable Conformer Generation, ICML 2024.
> [7] Abramson et al., Accurate structure prediction of biomolecular interactions with AlphaFold3, Nature 2024.
> [8] Yusuf et al., Review on Metal–Organic Framework Classification, Synthetic Approaches, and Influencing Factors: Applications in Energy, Drug Delivery, and Wastewater Treatment, ACS Omega 2022.

---

> > ### Comment · Reviewer_1rvy · 2024-11-27
> >
> > Thank you for your detailed responses. While most of my concerns have been addressed, I have a few follow-up questions:
> >
> > 1. Are the lattice parameters actually treated as global features of a finite 3D graph in the proposed framework? A simple "yes" is fine, as I understand that incorporating periodicity is a challenging issue.
> > 2. How is the rigid structure of each building block determined during the generation process?

---

> ### Author Response · Authors · 2024-11-27
>
> **Q1. Are the lattice parameters actually treated as global features of a finite 3D graph in the proposed framework? A simple "yes" is fine, as I understand that incorporating periodicity is a challenging issue.**
>
> Yes, they are treated as global features of a finite 3D graph. Thank you for understanding that incorporating periodicity is a significant challenge within our framework.
>
> **Q2. How is the rigid structure of each building block determined during the generation process?**
>
> Thank for you raising this question. We assume that the structure of each building block is known during the generation process. This is a valid assumption given the availability of extensive libraries of MOF building blocks with detailed structural information [1, 2]. For building blocks not found in these libraries, their 3D structure can be reliably determined using conformer generation tools, as MOFs typically consist of simple and rigid building blocks [3].
>
> We hope that our responses have addressed all your concerns thoroughly. If you have any further questions or require additional clarifications, we would be happy to provide them!
>
> ---
>
> [1] Boyd, Peter G., et al. "Data-driven design of metal–organic frameworks for wet flue gas CO2 capture." Nature 576.7786 (2019): 253-256.
> [2] Gibaldi, Marco, et al. "The HEALED SBU library of chemically realistic building blocks for construction of hypothetical metal–organic frameworks." ACS Applied Materials & Interfaces 14.38 (2022): 43372-43386.
> [3] Yusuf et al., Review on Metal–Organic Framework Classification, Synthetic Approaches, and Influencing Factors: Applications in Energy, Drug Delivery, and Wastewater Treatment, ACS Omega 2022.

---

> > ### Comment · Reviewer_1rvy · 2024-11-27
> >
> > Thanks for the answers! I would like to keep my origin score for acceptance.

---

> > > ### Author Response · Authors · 2024-11-27
> > >
> > > Thank you so much for your support!

---

### Official Review · Reviewer_6goZ · 2024-11-03

**Soundness:** 2
**Presentation:** 3
**Contribution:** 3
**Rating:** 6
**Confidence:** 4

**Summary:**

The paper presents MOFFLOW, a deep generative model aimed at predicting the structures of metal-organic frameworks (MOFs), leveraging a novel Riemannian flow matching framework. This approach models the building blocks of MOFs as rigid bodies, significantly reducing the complexity of structure prediction. MOFFLOW operates in the SE(3) space, effectively capturing the roto-translational dynamics of the metal nodes and organic linkers. The authors claim that MOFFLOW outperforms both conventional crystal structure prediction (CSP) methods and state-of-the-art deep learning models in terms of accuracy and scalability, handling unit cells with up to thousands of atoms.

**Strengths:**

1. The use of a Riemannian flow matching framework to handle the inherent complexity and modularity of MOFs is a novel aspect of the work. This approach allows MOFFLOW to reduce the dimensionality of the problem and improve scalability.
2. MOFFLOW's ability to handle large structures with thousands of atoms and maintain high accuracy is a significant advantage, setting it apart from conventional methods that struggle beyond smaller systems.
3. The authors provide detailed information on hyperparameters, training setup, and make their code and model checkpoints available, enhancing the paper's transparency and reproducibility.

**Weaknesses:**

1. While the paper compares MOFFLOW against CSP methods and the DiffCSP model, the absence of comparisons with more recent and potentially relevant MOF-specific deep learning models could limit the assessment of state-of-the-art competitiveness.
2.  Although MOFFLOW incorporates multiple technical innovations (e.g., block-level embedding updates, MOFAttention), the impact of these individual components is not deeply dissected through ablation studies. Such an analysis would strengthen the understanding of the contributions of each module.
3. While MOFFLOW is stated to be computationally efficient compared to baseline methods, a more detailed analysis of its training and inference costs relative to comparable models would help contextualize its practical deployment feasibility.
4. (Minor point) The paper mainly focuses on properties relevant to structure prediction accuracy (e.g., match rate and RMSE). It would be beneficial to explore how MOFFLOW's predictions perform in real-world settings that are essential for the practical use of MOFs.

**Questions:**

1. I would suggest the authors add necessary baselines for a fair comparison, which include but are not limited to CDVAE, MOFDiff, FlowMM, DiffCSP++, etc.
2. Providing insights into the relative contributions of different components (e.g., block-level update layers, MOFAttention module) would add clarity.
3. Including a detailed comparison of computational resources (e.g., GPU hours, memory usage) against other models would be valuable.

---

> ### Author Response · Authors · 2024-11-23
>
> Dear reviewer 6goZ,
>
> We deeply appreciate your thorough review and constructive comments on our manuscript. In the revised manuscript, we have marked the changes in **blue**. Below, we address the reviewer's concerns in detail.
>
> ---
>
> **W1/Q1. Absence of comparison with MOF-specific deep learning models. Add necessary (general CSP) baselines for a fair comparison, which include but are not limited to CDVAE, MOFDiff, FlowMM, DiffCSP++, etc.**
>
> First, we note that there exist no MOF-specific deep learning models that are directly comparable to ours. We already compare with the self-assembly algorithm of MOFDiff in Section 5.4. Note that MOFDiff solves a problem different to MOF structure prediction, so the algorithms are not directly comparable.
>
> To alleviate your concern, we have incorporated your suggestion and are currently adding the most recent CSP algorithms (FlowMM[1] and EquiCSP[2]) as new baselines. These experiments are ongoing, and results will be updated during the rebuttal period.
>
> In **Table A**, we report the performance of FlowMM evaluated after 8 days of training. We observe that FlowMM performs even worse than DiffCSP, which might be attributed to its high resource demands; training FlowMM for 70 epochs took 8 days on an 80GB A100 GPU, while DiffCSP completed 200 epochs in just 5 days on a 3090 GPU. Note that MOFFlow takes 5 days to train for the TimestepBatch version and 1~2 days for the Batch version (discussed extensively in W3/Q3). We are continuing to train FlowMM to ensure fair comparisons in the final paper.
>
> If the reviewer is aware of any more recent and relevant MOF-specific models appropriate for this task, we would be happy to evaluate them in our framework.
>
> ###### Table A. Structure prediction accuracy of Random Search (RS), Evolutionary Algorithm (EA), DiffCSP, FlowMM, and MOFFlow. MR denotes match rate and RMSE denotes root mean squared error.
> |                |                     |     stol = 0.5           |                           |     stol = 1.0          |                           |                        |
> |----------------|---------------------|--------------------------|---------------------------|-------------------------|---------------------------|------------------------|
> |                |     # of samples    |     MR (%)               |     RMSE                  |     MR (%)              |     RMSE                  |     Avg. time (s)      |
> |     RS         |     20              |     0.00                 |     -                     |     0.00                |     -                     |     332                |
> |     EA         |     20              |     0.00                 |     -                     |     0.00                |     -                     |     1959               |
> |     DiffCSP    |     1   /   5        |     0.09   /   0.34       |     0.3961   /   0.3848    |     23.12   /   38.94    |     0.8294  /    0.7937    |     5.37  /    26.85    |
> |     FlowMM*     |     1   /   5        |     0.00   /  0.00        |     -   /  -               |     0.6871  /   1.11     |     0.8487  /   0.7853     |     1.11  /   5.48      |
> |     EquiCSP*    |     1   /   5        |     0.00   /  0.00        |     -   /  -               |     0.9663 /   3.51     |     0.9663 /   0.9626  |     5.65  /   28.25     |
> |     MOFFlow    |     1   /   5        |     31.69  /    44.75     |     0.2820   /   0.2694    |     87.46   /   100.0    |     0.5183  /    0.4645    |     1.94  /   5.69      |
> *incomplete

---

> ### Author Response · Authors · 2024-11-23
>
> **W2/Q2. Lack of ablations to understand contributions on each module. Providing insights into the relative contributions of different components (e.g., block-level embedding, MOFAttention module) would add clarity.**
>
> Thank you for the suggestion. Based on your feedback, we performed ablation studies to evaluate the relative contributions of the block-level embedding module and the MOFAttention module. Specifically, we varied the number of layers and reduced the hidden dimension from 64 to 32, similar to the ablation studies conducted by DiffDock [3]. The results, summarized in **Table B**, indicate that a single MOFAttention layer has a greater impact on performance than a layer of the building block embedding module.
>
> ###### Table B. Ablation study on MOFAttention and building block embedding layers. We report the match rate (%) and RMSE of the base MOFFlow model and its variations with one layer removed. The results highlight that the MOFAttention layer has a greater impact on performance than a building block embedder layer.
> |                                          |     Match rate (%)    |     RMSE                |
> |------------------------------------------|-----------------------|-------------------------|
> |     Base                                 |     32.73             |     0.2743              |
> |     -1 MOFAttention   layer              |     30.59 (-2.14)     |     0.2785 (+0.0042)    |
> |     -1 building block embedder layer     |     31.37 (-1.36)     |     0.2749 (+0.0006)    |
>
> While it would be desirable to evaluate the relative contributions by completely removing each module, this is not possible since there exists no alternatives to our proposed pipeline.
>
> **W3/Q3. MOFFlow is stated to be computationally efficient compared to baseline methods. Including a detailed comparison of computational resources (e.g., GPU hours, memory usage, inference costs) against other models would be valuable.**
>
> Thank you for the insightful comment. We would like to clarify that MOFFlow’s computational efficiency is highlighted in terms of inference costs, as shown in Table 1. To address your concerns on training costs, we added a comparison of computational resources required to train each model in Appendix C. This covers batch size, GPU type and count, GPU memory usage, and total training time. We present a summary of this in **Table C** for your convenience.
>
> ###### Table C. Comparison of computational resources required for DiffCSP, FlowMM, and MOFFlow. We report the batch size, GPU type, number of GPUs, GPU memory utilization in GB, and training time in days/hours.
> |               |     Batch size     |     GPU Type (x number)            |     GPU memory utilization (GB)    |     Training time     (d: days, h: hours)    |
> |-----------|---------|---------|------------|-------------|
> |     DiffCSP                    |     8           |     24GB NVIDIA GeForce RTX 3090 (x1)    |     21.60 – 23.77                  |     5d 2h                                    |
> |     FlowMM*                     |     16        |     80GB NVIDIA A100-SXM4 (x1)           |     40.67 - 67.43                  |     8d 12h                                   |
> |     EquiCSP*                     |     8          |     80GB NVIDIA A100-SXM4 (x1)           |     -                  |     5d 2h                                   |
> |     MOFFlow (TimestepBatch)    |     160       |     24GB NVIDIA GeForce RTX 3090 (x8)    |     6.67 - 23.95                   |     5d 15h                                   |
> |     MOFFlow     (Batch)        |     160            |     24GB NVIDIA GeForce RTX 3090 (x8)    |     21.62 - 23.80                  |     1d 17h                                   |
> *incomplete
>
> In the process of addressing this suggestion, we identified that the TimestepBatch implementation [4] used in our main experiments takes longer time in terms of GPU hours. Motivated by this insight, we refactored our code, leading to the development of the Batch version ([link](https://anonymous.4open.science/r/MOFFlow-D0F2/README.md)), which requires significantly less training time to achieve similar performance (1~2 days vs. 5 days). Detailed discussions and results related to the Batch implementation have been added to Appendix D. Both versions will be released upon acceptance.
>
> |     Version         |     Training time (GPU hours)    |     Inference time    |     Match rate (%)    |     RMSE      |
> |---------------------|----------------------------------|-----------------------|-----------------------|---------------|
> |     TimestepBatch    |     1087.25                      |     1.94 s            |     31.69             |     0.2820    |
> |     Batch           |     332.74                       |     0.1932 s          |     32.73             |     0.2743    |
>
> We deeply appreciate the reviewer’s suggestion, which not only improved the completeness of our work but also guided us toward a meaningful improvement in our methodology.

---

> ### Author Response · Authors · 2024-11-23
>
> **W4. (Minor) Paper mainly focuses on properties relevant to structure prediction accuracy (e.g., match rate and RMSE). Explore how MOFFlow's predictions perform in real-world settings that are essential for the practical use of MOFs.**
>
> Thank you for the suggestion. MOFFlow is designed to be directly applicable to real-world workflows that support experimental synthesis. For instance, experimentalists can use MOFFlow to (1) predict a MOF structure from building blocks, (2) assess whether the predicted structure is desired, and (3) proceed with synthesis. This workflow parallels AlphaFold [5], saving the costs for experimental structure determination.
>
> ---
>
> ### References
>
> [1] Miller et al., FlowMM: Generating Materials with Riemannian Flow Matching., ICML 2024.
> [2] Lin et al., Equivariant Diffusion for Crystal Structure Prediction., ICML 2024.
> [3] Corso et al., DiffDock: Diffusion Steps, Twists, and Turns for Molecular Docking, ICLR 2023.
> [4] Yim et al., SE(3) diffusion model with application to protein backbone generation, ICML 2023.
> [5] Jumper et al., Highly accurate protein structure prediction with AlphaFold, Nature 2021.

---

> > ### Comment · Reviewer_6goZ · 2024-11-27
> > **Discussion**
> >
> > Thank you for addressing my concerns and adding additional results. This rebuttal is helpful. I have a few more questions that we may want to discuss a bit further:
> > - For the ablation studies, the gap seems small. Could you provide error bars for multiple runs to show the statistical significance?
> > - For the training costs, I noticed that only MOFFlow used multi-gpu training. Do the other baselines support this?
> > - Can you elaborate a bit more on the differences between those two "batched" implementations? What makes the improved version more efficient?

---

> ### Author Response · Authors · 2024-11-27
>
> **Q1. For the ablation studies, the gap seems small. Could you provide error bars for multiple runs to show the statistical significance?**
>
> Yes, we are currently running additional experiments to address your concerns and will keep our results updated.
>
> Please note that we initially aimed to produce statistically significant results by halving the number of layers in each module. However, this caused training to diverge, highlighting that both modules have important contributions.
>
> **Q2. Can you elaborate a bit more on the differences between those two "batched" implementations? What makes the improved version more efficient?**
>
> **TimestepBatch [1]:** This is the approach used in our original manuscript. To create a batch of size $B$, we sample a **single datapoint** from the dataset and generate **$B$ noised instances** of this single datapoint. The advantage of this method is that each batch contains the same number of building blocks and the same number of atoms within those building blocks, simplifying implementation and facilitating parallelization.
>
> **Batch:** This is how a usual batch is created for flow matching models. To create a batch of size $B$, we sample **$B$ datapoints** from the dataset and generate a **single noised instance** for each datapoint. While this method stabilizes training, it is much more difficult to implement, as it requires handling varying numbers of building blocks and differing numbers of atoms within those building blocks.
>
> The slow convergence of the TimestepBatch implementation arises because a batch effectively contains variations of a single data instance. This results in high gradient variability between each step, causing unstable training and longer convergence time.
>
> **Q3. For the training costs, I noticed that only MOFFlow used multi-gpu training. Do the other baselines support this?**
>
> Yes, the other baselines support multi-GPU training. DiffCSP was run on a single GPU since only one additional GPU was available at the time. FlowMM, on the other hand, requires an 80GB A100 GPU due to memory constraints. If you think this is a critical issue we will be happy to run DiffCSP again with 8 GPUs.
>
> We hope that our responses have addressed all your concerns thoroughly. If you have any further questions or require additional clarifications, we would be happy to provide them!

---

> > ### Comment · Reviewer_6goZ · 2024-11-27
> > **Discussion**
> >
> > Thanks for providing more details.
> >
> > - I'm a bit confused about why halving the number of layers would make the training diverge. You still have the necessary components/modules in your model but it's just the model is not that deeper as before. In deep learning, usually a deeper model would be harder to train to converge. That is why we have other tricks like residual connections. Can you provide more insights?
> > - I understood that "TimestepBatch" is easier to implement since everything is aligned and consistent. But for "batch", I think there's just one more step to do padding for building blocks and atoms. Is that right?
> > - It's okay to not run them under multi-gpu setting. I just want to understand the runtime comparison better. But one thing I would still want to ask is, since you have attention module, etc. involved, can you provide the big O complexity to help me understand the time complexity better in terms of both practice and theory?
> >
> > I'm also looking forward to the completed ablation studies.

---

> ### Author Response · Authors · 2024-11-29
>
> **Q. I'm a bit confused about why halving the number of layers would make the training diverge. You still have the necessary components/modules in your model but it's just the model is not that deeper as before. In deep learning, usually a deeper model would be harder to train to converge. That is why we have other tricks like residual connections. Can you provide more insights?**
>
> To be honest, it is challenging to pinpoint the exact reason, as understanding the behavior of deep learning models is still an open research question. One of our explanations is that halving the number of layers reduces the model’s capacity to a point where it becomes barely sufficient to fit the training data. Studies suggest that models in such cases are highly sensitive to even minor noise in the data, which can disrupt the model's global structure (refer to Discussion of Section 5 in [1]).
>
> Please note that we are using training stabilization techniques including gradient clipping, layer normalization, residual connections, and more.
>
> **Q. I understood that "TimestepBatch" is easier to implement since everything is aligned and consistent. But for "batch", I think there's just one more step to do padding for building blocks and atoms. Is that right?**
>
> We do not use padding for building blocks and atoms, as it would be highly memory-inefficient, especially given the large size and variability of MOF structures. In other words, the size of zero-padding is prohibitively large when a pair of small and large MOF structures are inside the same mini-batch.
>
> Instead, similar to existing GNN implementations [2], we represent each batch as a disconnected graph, with edges constructed between atoms within each building block and between building blocks within each data point. This inconsistent data structure posed some challenges during implementation.
>
> **Q. One thing I would still want to ask is, since you have an attention module, etc. involved, can you provide the big O complexity to help me understand the time complexity better in terms of both practice and theory?**
>
> We appreciate your question. We here share our analysis of the big O complexity.
>
> Consider a MOF containing $N$ atoms and $M$ building blocks, where the number of atoms in $m$th building block is denoted $N_m$ such that $\sum_{m=1}^M N_m = N$. Let $L$ denote the number of layers and $D$ the feature dimension.
>
> The big-O complexity of GNNs that operate on atomic coordinates is $O(L N^2 D)$. In contrast, our building block embedder has a time complexity of $O(L (N_1^2 + \dots+ N_M^2) D)$, as it operates on the building blocks. Note that $N_1^2 + \dots + N_M^2 \leq N^2$ since
> $$N^2 = (\sum_{i=1}^M N_m)^2 = \sum_{i=1}^N N_m^2 + 2 \sum_{m < n}N_m N_n \geq \sum_{i=1}^N N_m^2.$$
> The equality holds only when there is one building block, which does not occur in our dataset. Therefore, the time complexity of our building block approach is lower.
>
> We note that the time complexity of our attention module is $O(LM^2D)$, which is negligible since the number of building blocks is less than the number of atoms.
>
> For simplicity, this discussion assumes a fully connected graph for the upper-bound complexity. While we construct edges with radius cutoff in practice, analysis with this is difficult since the number of edges depends on the atomic positions.
>
> ---
>
> [1] Nakkiran, Preetum, et al. "Deep double descent: Where bigger models and more data hurt." Journal of Statistical Mechanics: Theory and Experiment 2021.12 (2021): 124003.
> [2] Fey, M., & Lenssen, J. E. (2019). Fast graph representation learning with PyTorch Geometric. arXiv preprint arXiv:1903.02428.

---

> ### Comment · Reviewer_6goZ · 2024-11-29
> **Discussion**
>
> Thank you for your reply. I think most of my concerns have been addressed. I'll raise the rating to 6. But I'm still looking forward to the ablation study.

---

> > ### Author Response · Authors · 2024-11-29
> >
> > Thank you for your support! We will share the results once the experiment is complete.

---

> ### Author Response · Authors · 2024-12-02
>
> Dear reviewer 6goZ,
>
> We appreciate your patience. We present ablation results over three runs, as shown below. In terms of match rate, the performance of removing a MOFAttention layer and a building block embedder layer is comparable. This indicates that both layers contribute similarly to overall performance.
>
> We attribute this to the complementary roles of these modules: the building block embedder captures atom-level resolution, while the MOFAttention layer focuses on block-level predictions. Together, they are crucial for achieving high performance.
>
> |                                          |     Match rate (%)    |     RMSE                 |
> |------------------------------------------|-----------------------|--------------------------|
> |     Base                                 |     32.71 ± 0.44      |     0.2752 ± 0.0014      |
> |     -1 MOFAttention   layer              |     31.31 ± 0.65      |     0.2773 ± 0.0018      |
> |     -1 Building block embedder layer     |     31.78 ± 0.63      |     0.2757 ± 0.0007      |
>
> We truly appreciate your helpful feedback and the effort you have put into improving our work.

---

> > ### Comment · Reviewer_6goZ · 2024-12-02
> > **Discussion**
> >
> > Thanks for providing these results. Although this comparison shows some evidence that these modules can help with the overall performance, it seems the numbers of different variants would overlap with the standard deviation. I guess this suggests that the proposed modules do not provide the most performance gain, which means without these modules, the model can still get a decent performance. However, this is a bit contradictory to the fact that reducing more layers would cause the model to diverge.
> >
> > I still have a bit of concern about this, but I'll keep my rating. I hope this more results and discussions can help AC form their own opinions.

---

> ### Author Response · Authors · 2024-12-04
>
> Dear reviewer 6goZ,
>
> We sincerely appreciate the time and effort you have dedicated to reviewing our work. We would like to address a possible misunderstanding in your comment: without the proposed modules, the pipeline is ill-defined and fails to define a valid mapping from MOF building blocks to the MOF structure.
>
> As noted earlier, removing additional layers caused the model to diverge, highlighting the essential role of these modules in achieving meaningful performance. To ensure stability, we limited our ablation to the removal of a single layer. While this resulted in relatively small changes, the differences remain meaningful; specifically, the difference between the base model and the removal of a MOFAttention layer does not overlap within the standard deviation. We hope this clarifies the importance of our proposed modules.

---

### Official Review · Reviewer_zVZp · 2024-11-03

**Soundness:** 3
**Presentation:** 3
**Contribution:** 3
**Rating:** 6
**Confidence:** 5

**Summary:**

The paper introduces MOFFlow, a frame-based flow matching approach to predict the structure of Metal Organic Frameworks (MOFs). MOFFlow treats the MOF building blocks (Metal nodes, organic links) as frames, and generates the 3D crystal structure by modeling the orientation and translation of the building blocks separately. A novel MOFAttention module is also introduced to model the hierarchical structure of MOFs. MOFFlow is benchmarked against DiffCSP, random structure search, and evolutionary algorithm baselines. The paper shows a significant improvement compared with DiffCSP in generating MOF structure, demonstrating the advantage of the hierarchical modeling approach in generating large MOF structures containing hundreds of atoms.

**Strengths:**

- The hierarchical modeling approach to separate the translation and orientation clearly demonstrates an advantage in predicting the structure of MOFs. The advantage is most clearly demonstrated in Fig. 6, indicating MOFFlow’s ability to scale to large structures with more than 200 atoms.
- The introduction of the MOFAttention module, inspired by the IPA module from AlphaFold 2, is interesting and novel.
- The authors show an improvement over the self-assemply algorithm from the MOFDiff paper. This demonstrates an advantage of learned methods over heuristic based methods.

**Weaknesses:**

- The individual pieces of the paper are not particularly novel. The dataset, evaluation, and the MOF decomposition method are similar to the MOFDiff work. The flow matching approach is similar to FrameFlow. The PCA axis to denote the orientations of the building block is similar to Gao & Gunnemann (2022).
- The paper lacks ablation study and/or more extensive benchmarking to illustrate the key component of their model that lead to the improvements. From DiffCSP to MOFFlow, there are many changes: 1) the adoption of a hierarchical modeling approach; 2) change from diffusion to flow matching; 3) change of the architecture of the score network. Which component leads to the biggest improvement? Although the likely answer is 1), more comparisons would greatly benefit the community.

**Questions:**

- How did the authors obtain the initial structure of the individual building blocks? Do they use the conformation from the data? Or predict those with a specific algorithm?
- Have the authors studied the number of steps required for the reverse process in flow matching?
- Have the authors studied the scalability of MOFFlow and the self-assembly algorithm used in MOFDiff, similar to Fig. 6?
- I would like to see more details on how Fig. 5 is produced. Since MOFDiff and MOFFlow models different types of problems, how do the authors perform a fair comparison in Fig. 5?

---

> ### Author Response · Authors · 2024-11-23
>
> Dear reviewer zVZp,
>
> We deeply appreciate your thorough review and constructive comments on our manuscript. In the revised manuscript, we have marked the changes in **blue**. Below, we address the reviewer's concerns in detail.
>
> ---
>
> **W1. The individual pieces of the paper are not particularly novel.**
>
> Thank you for your feedback. We respectfully highlight that our work introduces both novel pieces and a unique integration as meaningful contributions to the field. Specifically:
> 1. **MOFAttention module:** As noted by the reviewer, we introduce the MOFAttention module, a novel block-level update mechanism for MOFs.
> 2. **End-to-end framework:** We present the first end-to-end framework that jointly generates lattice parameters, block-level translations, and rotations, offering an comprehensive solution for MOF structure prediction.
> 3. **Novel integration**: By uniquely combining existing components with the MOFAttention module, we achieve a substantial performance improvement (31.69% vs. ~0%).
>
> **W2. Lack of ablation study that illustrates key component of the model. From DiffCSP to MOFFlow, there are many changes: 1) the adoption of a hierarchical modeling approach; 2) change from diffusion to flow matching; 3) change of the architecture of the score network.**
>
> Thank you for highlighting these points. Below, we provide ablation studies to evaluate the contributions of the building block embedder and MOFAttention, two core components of MOFFlow. Specifically, we varied the number of layers and reduced the hidden dimension from 64 to 32, similar to the ablation studies conducted by DiffDock [1]. The results, summarized in **Table A**, indicate that a single MOFAttention layer has a greater impact on performance than a layer of the building block embedding module.
>
> ###### Table A. Ablation study on MOFAttention and building block embedding layers. We report the match rate (%) and RMSE of the base MOFFlow model and its variations with one layer removed. The results highlight that the a MOFAttention layer has a greater impact on performance than a building block embedder layer.
>
> |                                          |     Match rate (%)    |     RMSE                |
> |------------------------------------------|-----------------------|-------------------------|
> |     Base                                 |     32.73             |     0.2743              |
> |     -1 MOFAttention   layer              |     30.59 (-2.14)     |     0.2785 (+0.0042)    |
> |     -1 building block embedder layer     |     31.37 (-1.36)     |     0.2749 (+0.0006)    |
>
> Regarding the specific ablations suggested by the reviewer:
> 1. **Hierarchical vs. non-hierarchical.** We are currently training FlowMM as an additional baseline, which can be considered the non-hierarchical counterpart of MOFFlow.
> 2. **Diffusion vs. flow matching.** Implementing a diffusion-based version of MOFFlow is challenging due to our specific choice of priors, which requires deriving an appropriate closed-form kernel for diffusion that converges to this prior. While this task is infeasible within the current timeline, we will explore its implementation and aim to share results in the final version of our paper.
> 3. **Other network architectures.** Since our work is the first to introduce a block-level architecture for MOFs, direct comparisons with other architectures are difficult. Instead, we focus on evaluating relative contributions of key model components, as shown in **Table A**.

---

> ### Author Response · Authors · 2024-11-23
>
> **Q1. How did the authors obtain the initial structure of the individual building blocks? Do they use the conformation from the data? Or predict those with a specific algorithm?**
>
> As explained in Section 5.2 (line 367-369), we use the metal-oxo decomposition algorithm of MOFid [2] to obtain the initial building block structures. This algorithm takes a complete MOF structure (in CIF format) as input, identifies the metal-oxo clusters (metal nodes), and classifies the remaining fragments as organic building blocks.
>
> **Q2. Have the authors studied the number of steps required for the reverse process in flow matching?**
>
> Thank you for pointing this out. Yes, we studied the number of steps to balance performance (match rate, RMSE) and sampling time, concluding that 50 integration steps offered the best trade-off, and used it in our main experiments. These results are now included in Appendix G (and cross-referenced in line 425) of the revised manuscript.
>
> **Q3. Have the authors studied the scalability of MOFFlow and the self-assembly algorithm used in MOFDiff, similar to Fig. 6?**
>
> We appreciate your suggestion. To address your concern, we ran additional analysis and added Appendix H in our revised manuscript, which studies the scalability and sampling efficiency of MOFFlow and the self-assembly algorithm. In summary, while MOFFlow demonstrates superior overall performance and significantly faster inference, the self-assembly algorithm exhibits slightly better scalability for structures with many building blocks.
>
> **Q4. I would like to see more details on how Fig. 5 is produced. Since MOFDiff and MOFFlow models different types of problems, how do the authors perform a fair comparison in Fig. 5?**
>
> We note that Figure 5 compares MOFFlow to DiffCSP, not MOFDiff. Indeed, as the reviewer mentions, we do not compare with MOFDiff since it does not address the structure prediction problem.
>
> ---
>
> ### Reference
> [1] Corso et al., DiffDock: Diffusion Steps, Twists, and Turns for Molecular Docking, ICLR 2023.
> [2] Bucior et al., Identification schemes for metal-organic frameworks to enable rapid sesarch and cheminformatic analysis, 2019.

---

> > ### Comment · Reviewer_zVZp · 2024-11-24
> >
> > I appeciate the authors for their detailed response and additional experiments to address my comments. I've read their response as well as the comments from other reviewers. I will keep my current score.
> >
> > Additional comments based on authors' responses to address in camera ready version.
> >
> > - For Q1, I understand that MOFid is used to decompose the MOF structure to building blocks. I am asking how the authors obtain the initial 3D structure of the building blocks. Do the authors take average overing 3D structures in training data? Or do they use RDKit?
> > - For Q2, why does the model archieve best performance at 50 steps, instead monotonously increasing performance with more integration steps?

---

> ### Author Response · Authors · 2024-11-27
>
> Thank you for your support! Regarding the follow-up questions:
>
> **Q1. I understand that MOFid is used to decompose the MOF structure to building blocks. I am asking how the authors obtain the initial 3D structure of the building blocks. Do the authors take average over 3D structures in training data? Or do they use RDKit?**
>
> We apologize for misunderstanding your question. With ~2 million building blocks in the training data, categorizing them to compute an average 3D structure is computationally infeasible. Instead, we follow MOFDiff and treat each building block as a unique entity, allowing the building block embedder to account for discrepancies in structure by learning consistent representations during training. During generation, we assume the structures are known; this is a valid assumption given the inherent rigidity of building blocks [1] and the availability of extensive libraries of MOF building blocks with detailed structural information [2, 3].
>
> We also find your idea of using RDKit compelling, as it could improve robustness and enable text-based MOF structure prediction. Thank you for sharing this valuable idea.
>
> **Q2. Why does the model achieve the best performance at 50 steps, instead of monotonically increasing performance with more integration steps?**
>
> This is an interesting observation and we share your curiosity. FlowMM, which exhibits a similar trend, also lacks a clear explanation for this behavior. One of our thoughts is that increasing the number of integration steps may lead to a greater accumulation of numerical errors, which could counteract the benefits of finer integration. However, further investigation is needed to determine the exact reason for this behavior.
>
> We hope that our responses have addressed all your concerns thoroughly. If you have any further questions or require additional clarifications, we would be happy to provide them!
>
> ---
>
> [1] Yusuf et al., Review on Metal–Organic Framework Classification, Synthetic Approaches, and Influencing Factors: Applications in Energy, Drug Delivery, and Wastewater Treatment, ACS Omega 2022.
> [2] Boyd, Peter G., et al. "Data-driven design of metal–organic frameworks for wet flue gas CO2 capture." Nature 576.7786 (2019): 253-256.
> [3] Gibaldi, Marco, et al. "The HEALED SBU library of chemically realistic building blocks for construction of hypothetical metal–organic frameworks." ACS Applied Materials & Interfaces 14.38 (2022): 43372-43386.

---

### Official Review · Reviewer_hN1H · 2024-11-04

**Soundness:** 4
**Presentation:** 4
**Contribution:** 3
**Rating:** 8
**Confidence:** 4

**Summary:**

The authors tackle the problem of crystal structure prediction (CSP) of Metal-Organic Frameworks. Typically, CSP is performed using ab initio calculations using density functional theory (DFT). However, DFT is computationally expensive and scales poorly for large complex systems such as MOFs. MOFFLow presents a MOF-specific generative model for CSP that operates on secondary building units rather than isolated atoms for structure prediction.

**Strengths:**

* Outperforms existing deep learning-based structure prediction models by a significant amount.
* Highly scalable.

**Weaknesses:**

* Limited applicability on domains.
* Crystal structures that cannot be decomposed into secondary structures cannot readily use these techniques

**Questions:**

* Awkward notation in equation 1. L is 3x3. k is 3 x 1. Lk is therefore 3x1. x_n in line 143 is  1 x 3. So, x_n + Lk is not proper.
* Line 156: building blocks contain a set of “local” coordinates. What is the reference frame used for these coordinates?
* Section 5.2: Are the property evaluations done on predicted structures and compared with property evaluations on ground-truth structures?

---

> ### Author Response · Authors · 2024-11-23
>
> Dear reviewer hN1H,
>
> We deeply appreciate your thorough review and constructive comments on our manuscript. In the revised manuscript, we have marked the changes in **blue**. Below, we address the reviewer's concerns in detail.
>
> ---
>
> **W. Limited applicability. Crystal structures that cannot be decomposed into secondary structures cannot readily use these techniques.**
>
> We agree with this point. Nevertheless, we still believe that our work is significant since decomposable crystals like MOFs, covalent organic frameworks, and molecular crystals, have important real-world applications such as gas separation[1], catalysis[2], drug delivery[3], and sensing[4].
>
> ---
>
> **Q1. Awkward notation in equation 1. $x_n + Lk$ is not proper.**
>
> We apologize for the oversight. We have revised our manuscript to correct this as $x_n' =x_n + kL^{\top}$ where $k =(k_1, k_2, k_3) \in \mathbb{Z}^{1\times3}$ and $L = (l_1, l_2, l_3) \in \mathbb{R}^{3 \times 3}$. We appreciate your careful attention to detail.
>
> **Q2. What is the reference frame for local building block coordinates in line 156?**
>
> The reference frame is the PCA axes, with their orientation fixed by a reference equivariant vector as outlined in [5]. This is detailed in Section 4.1, and we have added a cross-reference at line 156 to improve readability.
>
> **Q3. In Section 5.2, are the property evaluations done on the predicted structures and compared with the property evaluations on ground-truth structures?**
>
> Yes, we compare the property evaluations on predicted structures to that of ground-truth structures. We apologize for the ambiguity and have revised Section 5.2 (lines 461-467) to clarify this.
>
> ---
>
> ### References
> [1] Li et al., Recent advances in gas storage and separation using metal-organic frameworks, Materials Today 2018.
> [2] Lee et al., Metal-organic framework materials as catalysts, Chemical Society Reviews 2009.
> [3] Horcajada et al., Metal-organic frameworks in biomedicine, Chemical Reviews 2012.
> [4] Kreno et al., Metal-organic framework materials as chemical sensors, Chemical Reviews 2012.
> [5] Gao & Günneman, Ab-initio potential energy surfaces by pairing gnns with neural wave functions, ICLR 2022.

---

> > ### Comment · Reviewer_hN1H · 2024-11-26
> >
> > Thank you to the detailed updates and fixes. As this works presents a strong improvement on crystal structure prediction for decomposable structures, I will keep my current score.

---

> > > ### Author Response · Authors · 2024-11-27
> > >
> > > We appreciate your positive feedback and are happy to address any further questions!

---

### Meta-Review · Area_Chair_Wpj1 · 2024-12-19

**Metareview:**

The submission proposes a method for generating the 3D structure of Metal-Organic Framework systems (MOFs). The major contribution is a hierarchical modeling of the structures to reduce the cost for handling a large number of atoms. The method is based on Riemannian flow matching on the SE(3) space to account for geometric invariance of the target distribution, which is sound enough.

All reviewers appreciated the soundness and the improved performance and effectiveness in handling large systems (relative to typical crystalline materials). Performance comparison with a related work MOFDiff is also informative. Reviewers also raised a few insufficiencies:
* Investigation of the effect of each technical modification (raised by Reviewer zVZp and 6goZ). In the rebuttal, the authors provided results for ablating the MOFAttention layer, which serve as a positive evidence. Regarding this, Reviewer 6goZ raised a follow-up complaint that the improvement in the ablation study with one layer removed is insignificant, while removing more layers makes the model to diverge. Although the reviewer did not make further responses, I suppose the authors' further reply makes sense (it could be the case that removing more layers shatters the model).
* Reviewer 6goZ also asked for comparison with more recent relevant baselines. In the rebuttal, the authors made further efforts in comparisons with a bit broader related methods, though the results are not trained to convergence for making any conclusion.
* Reviewer 1rvy asked for presenting more details of periodicity handling, a more holistic overview of the architecture, and limitations of the rigid body assumption. Authors seem to have sufficiently addressed these points in rebuttal.

As such, reviewers are mostly satisfied regarding the raised insufficiences, and I did not find more major problems. I hence recommend accept. To make the paper stronger, I hope the authors could continue on investigating the effect of hierarchical modeling and using flow matching, and further improve presentation by providing the promised details (also include proper structure initialization method).

**Additional Comments On Reviewer Discussion:**

Reviewer 6goZ raised his/her score to positive after the rebuttal. The authors provided further ablation study results and more baseline comparison results, which covers the reviewer's concerns.

---

### Decision · Program_Chairs · 2025-01-22

Accept (Poster)